# Whom to Query for What: Adaptive Group Elicitation via Multi-Turn LLM Interactions

Ruomeng Ding [* 1] Tianwei Gao [* 1] Thomas P. Zollo [2] Eitan Bachmat [3] Richard Zemel [2] Zhun Deng [1]

## Abstract

Eliciting information to reduce uncertainty about latent group-level properties is a central problem in collective assessment, preference modeling, and opinion aggregation, and is especially important in survey-based studies. While natural language interactions provide a flexible interface, existing methods typically rely on fixed questionnaires and static respondent sets, and do not adapt to partial or missing responses across rounds. To address this gap, we study adaptive information elicitation through multi-turn interactions between a large language model and a group of individuals, where both queries and respondents are adaptively selected to infer latent group properties. We propose a theoretically grounded framework that, at each round, jointly selects a query and a subset of respondents based on previously observed responses to efficiently reduce uncertainty about a target latent quantity (e.g., group-level political inclination). Motivated by practical survey constraints, such as limited questions and costly participation, our strategy maximizes information gain under a fixed budget. To handle missing and incomplete responses, we combine graph neural networks for aggregating/imputing partial group information with an information-theoretic criterion that guides per-round selection. Across three real-world opinion datasets, we achieve consistent improvements in population-level response prediction under constrained budgets, including over a 12% relative gain on CES at a 10% respondent budget. Code is available at: https://github.com/ZDCSlab/Group-Adaptive-Elicitation.

---
*Equal contribution. Authors are listed in alphabetical order. [1] University of North Carolina at Chapel Hill, Chapel Hill, NC, USA [2] Columbia University, New York, NY, USA [3] Ben-Gurion University of the Negev, Beersheba, Israel . Correspondence to: Zhun Deng <zhundeng@cs.unc.edu>.

*Proceedings of the 43rd International Conference on Machine Learning*, Seoul, South Korea. PMLR 306, 2026. Copyright 2026 by the author(s).

## 1. Introduction

Surveys and other collective assessments work by asking a limited set of questions to a subset of the population of interest in order to infer latent population properties, for example, county-level political inclination, student skill profiles, or employee preferences over policy changes. Because each additional question and each completed response incurs real cost, including respondent burden, interviewer time, and participation incentives, instruments are typically kept short, and modern survey practice increasingly emphasizes strategic effort allocation to manage the cost error tradeoff (Dillman et al., 1993; Groves & Heeringa, 2006; Chun et al., 2018). These constraints often result in missing data and breakoffs, so downstream estimates in policy and the social sciences are frequently based on sparse, partially observed responses (Meyer et al., 2015).

Large language models (LLMs) offer a natural interface for making these processes adaptive: they can pose questions in natural language, condition on interaction history, and produce predictive distributions over responses. Recent work has begun using autoregressive models as tools for adaptive elicitation, selecting questions to maximize expected information gain from observed histories (Wang et al., 2025). However, most existing approaches primarily optimize *what to ask* under an implicit fixed respondent pool, while the dominant bottleneck in many deployments is the number of completed responses that can be collected; improving efficiency therefore also requires deciding *who to ask*, and leveraging population structure (e.g., demographics and similarity across individuals) so that information gathered from a few respondents generalizes to the broader group.

To address these challenges, we study adaptive group elicitation: a multi-round interaction in which a central agent jointly selects the next question to ask and a subset of respondents to query in order to reduce uncertainty about a latent group quantity under a fixed budget. In an election survey, this means deciding not only which policy question to ask next, but also which voters to contact when only a small fraction can be interviewed each round. Our approach couples LLM-based predictive inference with population-level propagation: a meta-trained LLM scores candidate questions by expected information gain given the current interaction

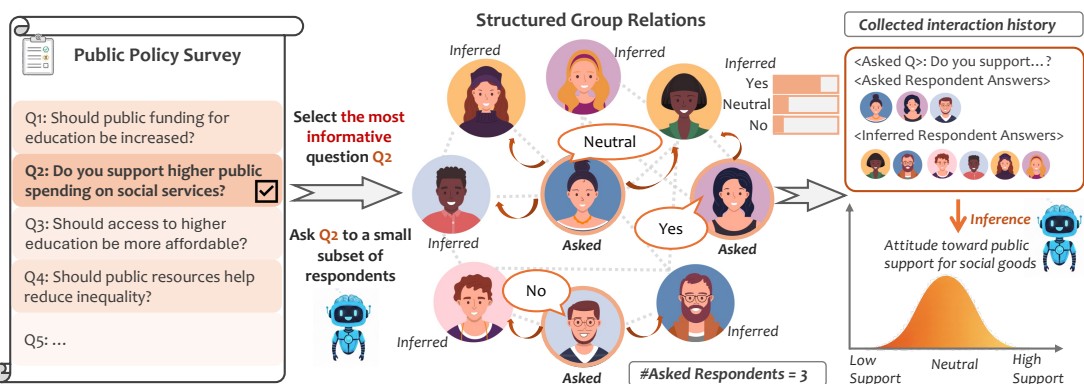

*Figure 1.* Overview of group adaptive elicitation. The method jointly selects informative queries and leverages structured group relations, asking only a few respondents while inferring remaining responses to learn a shared latent group preference.

history, while a heterogeneous graph neural network (GNN) aggregates observed responses and demographics to impute missing responses and refine representations of similarity across respondents (Suh et al., 2025). The resulting loop uses these graph-informed representations to select a diverse, representative respondent subset each round, and updates the graph with new observations before repeating.

**Our contribution.** We (i) formalize adaptive group elicitation as a joint decision problem over questions and respondents under query and participation budgets, (ii) propose a practical instantiation that combines an LLM-based expected-information-gain objective with heterogeneous GNN propagation for imputation and respondent selection, and (iii) provide theoretical justification via a predictive (de Finetti–style) perspective for graph-structured data and near-optimal guarantees for greedy joint selection under standard assumptions. We then evaluate on three real-world opinion datasets, showing consistent gains under constrained budgets (including a $> 12\%$ relative improvement on CES at a $10\%$ respondent budget). The remainder of the paper presents the setup and framework, develops the theory, and reports experimental results and ablations.

## 2. Preliminaries

We introduce several key components that provide the theoretical and practical foundations of our framework.

### 2.1. De Finetti's Theorems and Predictive Inference

The theoretical foundation of our approach builds on de Finetti's predictive perspective, which connects uncertainty quantification over a latent entity with forecasting the future behavior of the induced observables.

Specifically, a generalized version of de Finetti's theorem (see Fact 2.1) shows, for a sequence of random variables $\{Y_n\}_{n=1}^{\infty}$, denote the one-step-ahead predictive distribution

$p_t(y|Y_{1:t}) := p(Y_{t+1} \mid Y_{1:t})$, one may expect the existence of a latent entity $U$ that governs future behavioral patterns, in the sense that all uncertainty about future observations arises from uncertainty about $U$, that is there exists a space $\mathcal{U}$ with a measure $\mu$, such that

$$p(Y_{1:n}) = \int_{\mathcal{U}} \prod_{i=1}^{n} p(Y_i|U)\mu(dU),$$

Building on this perspective, one can apply a recursive, autoregressive update of beliefs about $U$, enabling principled uncertainty quantification and adaptive elicitation at the individual level.

**Fact 2.1** (Informal, Generalized De Finetti's Theorem (Berti et al., 2004; Fong et al., 2024)). *For a sequence of random variables $\{Y_t\}_{n=1}^{\infty}$ taking values in a Polish space $E$, under certain martingale condition on $\{p_t\}_{t=1}^{\infty}$, $p_n \rightarrow p_{\infty}$ for some limiting measure $p_{\infty}$ and we can recover the latent entity $U$ from the limiting measure $p_{\infty}$.*

We will further discuss about this assumption and justification about Fact 2.1 in our framework in Section 4 and Section A.2.

### 2.2. Heterogeneous GNN for Group Simulation

Modeling relational structure in group settings is challenging due to partial observations and complex dependencies among individuals. Heterogeneous graph neural networks provide a flexible framework by capturing multiple node types and diverse relational patterns. Suh et al. (2025) propose a heterogeneous graph $\mathcal{G}_{\text{heter}}$ with member, query, and demographic nodes, where edges encode demographic links and observed responses, framing response prediction as a link prediction problem and achieving strong empirical performance. Notably, their GNN model can match or even surpass strong LLM baselines despite being three orders of magnitude smaller. Message passing over this structure

enables robust prediction and imputation under sparse observations. This relational modeling component forms an important foundation that we adapt and incorporate into our framework in Section 3.

## 2.3. Notation

For clarity, we introduce generic notation that applies across all application settings. Let $G = (\mathcal{V}, \mathcal{E})$ denote a graph with node set $\mathcal{V}$ and edge set $\mathcal{E}$. For a node $v \in \mathcal{V}$, $N(v)$ represents the set of its neighbors, and $S(v) = \mathcal{V} \setminus (N(v) \cup \{v\})$ is the set of nodes not adjacent to $v$. We let $\mathcal{U}$ denote a generic space of latent entities that are the target of inference, $\mathcal{X}$ a set of candidate queries, and $\mathcal{Y}$ the space of possible responses. When $\mathcal{V}$ represents a group of members, for any subset of members $V \subset \mathcal{V}$ and a query $x \in \mathcal{X}$, we denote by $Y^V$ the vector of responses from members in $V$.

For a random variable $Z$, we define its entropy to be $H(Z) = -\int_z p(z) \log p(z) dz$, where $p$ is the probability density function of $Z$. Accordingly, we can follow the standard definition of conditional entropy and obtain the conditional entropy of $Z_1$ with respect to $Z_2$ as $H(Z_1|Z_2) = -\int_{z_2} \int_{z_1} p(z_1, z_2) \log p(z_1|z_2) dz_1 dz_2$.

## 3. Group Adaptive Elicitation Framework

As discussed in the introduction, many real world applications aim to uncover group latent properties through limited interactions. The resulting data is typically sparse, both in the number of queries posed and in the responses obtained. For instance, political surveys can administer only a small set of key questions under constraints of time, cost, and privacy, and responses are often collected from only a subset of voters due to budget limits and nonparticipation.

**Overview of our mechanism.** To address limited query budgets and missing responses in the interactive scenarios described above, we propose a framework that iteratively reduces group level uncertainty by combining individual predictive modeling with relational imputation. At each iteration, the framework adaptively selects both questions and a subset of group members based on the interaction history, inferred latent entities, and learned relational structure. This joint selection enables informative queries to be directed towards representative respondents to maximize expected information gain about the underlying group property.

Our approach is guided by two principles: ❶ quantifying uncertainty for each group member by aggregating interaction history, including imputed responses, and ❷ leveraging the group's relational structure to share information and recover missing data. Together, these principles give rise to a two-stage process at each interaction round:

1. **Individual uncertainty quantification:** We maintain a

predictive model for each group member with latent entity $U$ given the interaction history $\mathcal{H}_{t-1}$. Our overall uncertainty about the group is computed as the sum of individual uncertainties. This is achieved by meta-training an LLM on interaction history to accurately predict individual responses.

2. **Adaptive group interaction strategy:** At each round $t$, we select the next question $X_t \in \mathcal{X}$ and a subgroup of respondents $R_t \subset \mathcal{V}$ to query. The selection aims to (approximately) maximize the reduction in overall group uncertainty. Crucially, after observing new responses, we leverage a heterogeneous GNN that propagates information across the group's relational graph. This allows us to impute responses from group members not being queried, making the aggregation of individual uncertainties increasingly informed and efficient. We provide theoretical justification for our elicitation strategy in Section 4.

The proposed formulation enables computationally tractable adaptive group elicitation by combining individual level inference with relational reasoning to reduce group level uncertainty under sparse query responses. Below, we formally present our framework for efficiently characterizing group latent entities.

### 3.1. Pretraining

In the pretraining phase, our framework includes two components: a predictive language model for quantifying individual uncertainty and a heterogeneous GNN for leveraging relational structure. The language model predicts individual responses from interaction history and predefined features determined by the dataset, which may include demographic information in political surveys or other relevant attributes depending on the application context, while the GNN propagates information across members to impute missing answers and reduce uncertainty about the group latent entity.

**Training data.** To maintain consistency with the notation used in our later GNN formulation, we denote a group of $n$ members as $\mathcal{V} = \{v_i\}_{i=1}^n$, where $i$ indexes individuals. By observing query-response sequences over $T$ interaction rounds, we obtain a training dataset $\mathcal{D}_{\text{train}} = \{(X_{1:T}^{v_i}, Y_{1:T}^{v_i})\}_{i=1}^n$, where $(X_{1:T}^{v_i}, Y_{1:T}^{v_i})$ denotes the query-response sequence for the $i$-th group member.

**LLM for prediction inference.** We first meta-train a language model to perform predictive inference at the individual level, avoiding the need for a parametric prior over the complex latent space $\mathcal{U}$, as in (Ye & Namkoong, 2024; Wang et al., 2025). We finetune the parameters $\theta$ of a LLM to maximize the likelihood of autoregressive prediction:

$$\hat{\theta} = \arg\max_{\theta} \sum_{i=1}^{n} \sum_{t=1}^{T-1} \log p_{\theta}(Y_{t+1}^{v_i}|X_{t+1}^{v_i}, \mathcal{H}_t^{v_i}) \quad (1)$$

where $\mathcal{H}^v$ denotes the interaction history collected from group member $v$. As in (Ye & Namkoong, 2024; Wang et al., 2025), the above objective trains the LLM to serve as a robust predictor of individual group member behavior, enabling reliable uncertainty quantification and determining the informativeness of queries with respect to the latent entity given each group member's interaction history. This predictive capability allows the LLM to estimate the expected information gain of candidate queries, which is essential for guiding adaptive query selection in our framework.

**Heterogeneous GNN for group simulation.** The LLM's predictive inference benefits from complete interaction histories for each group member; however, in our setting the observed query-response data can be sparse in both the number of queries and the subset of respondents. To impute missing responses and thereby bolster the LLM's ability to characterize uncertainty over the latent entity, we employ a heterogeneous GNN. The GNN is well suited for this setting, as it propagates information across the group to perform robust imputation that compensates for partial observations.

Inspired by the approach in (Suh et al., 2025), we model the problem of response prediction for group members as link prediction in a heterogeneous graph with relational GNN. Specifically, we construct an extended version of graph $\mathcal{G}_{\text{heter}} = (\mathcal{V}_{\text{heter}}, \mathcal{E}_{\text{heter}})$ that contains three types of nodes corresponding to group members $\mathcal{V}$, demographic features $\mathcal{V}_f$, and choices $\mathcal{V}_c$. The edge set $\mathcal{E}_{\text{heter}} = \mathcal{E}_f \cup \mathcal{E}_c$ contains two subsets of edges of different types, the edge set $\mathcal{E}_f$ connects members to their features and response edge set $\mathcal{E}_c$ connects members to their selected choices.

Specifically, based on the training dataset $\mathcal{D}_{\text{train}}$ and query set $\mathcal{X}$, we construct the graph as follows. The node subsets and the edges between different types of nodes are given by: ❶ *Group member nodes $\mathcal{V}$*: We associate each group member in $\mathcal{V}$ with a node in $\mathcal{V}_{\text{heter}}$ and identify the group $\mathcal{V}$ in the training dataset with the associated node subset in $\mathcal{G}_{\text{heter}}$. ❷ *Feature nodes*: Each node represent an individual feature. For instance in political survey, we disretize demographic features such as age, gender and education level into categorical bins, such as "Age: 18-29", each bin corresponds to a feature node. A member node connects to all matching feature nodes. ❸ *Query-choice nodes $\mathcal{V}_c$*: Each node represents a unique (query $x$, choice $c$) pair. A member is connected to all the query-choice nodes (query $x$, choice $c$) such that it selects choice $c$ for query $x$.

The GNN is trained to learn the feature and query-choice nodes embeddings and relation-specific message passing mechanism using link prediction objective. Let $d$ be the embedding dimension, both the features and query-choice nodes embeddings are initiated with $d$-dimensional learnable embeddings and the embeddings of member nodes are initiated with uniform vector $e := (1, \ldots, 1) \in \mathbb{R}^d$. We

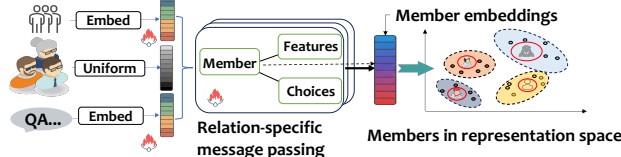

*(a)* Embedding-based clustering and subgroup selection

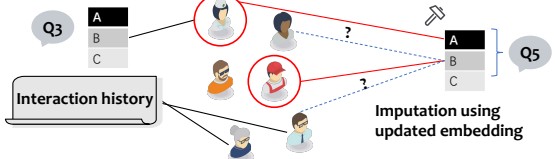

*(b)* Imputation using updated embedding

*Figure 2.* Overview of Heterogeneous GNN-based subgroup of respondents selection and imputation.

construct the training objective as follows. First, we randomly mask a subset of member-choice edges $\mathcal{E}_{\text{mask}}$. Then for each member $v$ and any query $q$ with its set of choices $\mathcal{C}_q$, the probability that $v$ selects choice $c \in \mathcal{C}_q$ is:

$$p(c \mid v, q) = \frac{\exp(\langle h_v, h_c \rangle / \tau)}{\sum_{c' \in \mathcal{C}_q} \exp(\langle h_v, h_{c'} \rangle / \tau)}, \qquad (2)$$

where $h_v$ and $h_c$ are the final-layer embedding vectors obtained by the GNN using message passing on the partially masked graph, $\langle \cdot, \cdot \rangle$ is the dot product, and $\tau$ is a learnable temperature parameter. The training optimization objective is to minimize the cross entropy loss for the masked edges:

$$\min - \sum_{(v,c) \in \mathcal{E}_{\text{mask}}} \log p(c \mid v, q(c)).$$

where $q(c)$ is the query corresponding to choice $c$.

### 3.2. Testing-Time Adaptive Inference

At test time, we are given a new group of members $\mathcal{V}^{\text{test}}$, a candidate query set $\mathcal{X}_c \subset \mathcal{X}$, and a held-out evaluation set $\mathcal{X}_h \subset \mathcal{X} \setminus \mathcal{X}_c$. Responses to questions in $\mathcal{X}_h$ serve as proxies for the latent entity of interest. We construct the heterogeneous graph for the testing group $\mathcal{V}^{test}$ following the same procedure as in the training phase with the same set of query-choice nodes and feature nodes as defined during training. Initially, when no query-response data is observed, we connect the test group members to feature nodes based on their demographic attributes. We initialize feature and query choice nodes with learned embeddings and member nodes uniformly. By integrating the pre-trained LLM and heterogeneous GNN, we iteratively select questions and respondents to reduce uncertainty about the group latent entity. Performance is evaluated by prediction accuracy on held-out queries in $\mathcal{X}_h$ across all members, enabling robust inference under sparse interactions.

**Adaptive query selection.** At each round $t$, given the interaction history $\mathcal{H}_{t-1}$, we select both a question $X_t$ and a subset of respondents $R_t \subset \mathcal{V}^{\text{test}}$ that maximize the expected information gain (EIG) about the group's latent entity. First, for a member $v$, to quantify uncertainty about the latent entity $U_v$, we adopt de Finetti's predictive perspective in (Wang et al., 2025), which used the conditional entropy over future observations as a proxy. Specifically, we define the uncertainty as:

$$H(U_v|\mathcal{H}_{t-1}^v) := \sum_{x \in \mathcal{X}_h} H(Y_x^v|X = x, \mathcal{H}_{t-1}^v), \quad (3)$$

where the entropy is computed over the held-out evaluation set $\mathcal{X}_h$. This provides a practical measure of the confidence with which we can predict the group member $v$'s responses to diagnostic questions. We then compute EIG using the pre-trained LLM $p_{\hat{\theta}}$ as the sum of individual-level information gains:

$$\text{EIG}(x; \mathcal{H}_{t-1}) = \sum_{v \in \mathcal{V}^{\text{test}}} \left( H(U_v|\mathcal{H}_{t-1}^v) - \mathbb{E}_{Y_t^v}[H(U_v|\hat{\mathcal{H}}_t^v)] \right),$$
$$(4)$$

where $\hat{Y}_t^v \sim p_{\hat{\theta}}(Y_t^v|X_t = x, \mathcal{H}_{t-1}^v)$ is the simulated response of member $v$ to query $x$, and $\hat{\mathcal{H}}_t^v = \mathcal{H}_{t-1}^v \cup \{(x, \hat{Y}_t^v)\}$ is the updated history. The EIG is a widely used criterion that assesses how much a query-response pair reduces uncertainty about the latent entity, thus, we select the query that maximizes the group EIG:

$$X_t = \arg \max_{x \in \mathcal{X}_c \setminus \{X_i\}_{i=1}^{t-1}} \text{EIG}(x; \mathcal{H}_{t-1}). \quad (5)$$

**Group-relational respondents selection.** To select a representative subgroup of respondents, we leverage relational structure using embeddings from the heterogeneous GNN. Let $h_v$ denote the final layer embedding for member $v \in \mathcal{V}^{\text{test}}$. Since similar embeddings imply similar response patterns, we select a diverse subgroup by maximizing coverage of the embedding space. Given a budget $k$, we cluster members based on these embeddings and choose the $k$ cluster centers as the representative subgroup $R_t$.

**Message propagation for imputation and belief updating.** After collecting responses $Y^{R_t}$ to query $X_t$ from the selected subgroup $R_t$, we update the heterogeneous graph with the new observations and perform message passing using the pre-trained GNN to refine node embeddings. These embeddings are used to impute unobserved responses for unqueried members, which are then added to the interaction history to form $\mathcal{H}_t$. The enriched graph and updated history enable improved response imputation and uncertainty quantification in subsequent rounds, thereby progressively reducing uncertainty about the group's latent entity. We defer details about imputation and belief updating to Appendix B.

### 3.3. Computational Cost Analysis.

Our method has comparable inference cost to Meta-Greedy with negligible GNN overhead. The dominant cost comes from LLM-based expected information gain (EIG) estimation, which requires sampling simulated future responses to evaluate uncertainty reduction, with complexity $O(|V||X_c||X_h|N)$, where $|V|$ is the test-time group size, $X_c$ and $X_h$ denote candidate and held-out query sets, and $N$ is the number of Monte Carlo samples. This LLM sampling bottleneck is shared by all strategic query-selection baselines, while random selection avoids this cost at the expense of downstream performance. Preference imputation is formulated as message passing on a heterogeneous graph over group members, demographic features, and query items, with complexity $O(|V| + |V_f| + |V_c| + |E_f| + |E_c|)$. The graph construction and message passing are linear in the number of nodes and edges, allowing the imputation module to scale efficiently with increasing group size. Moreover, because the query budget is typically much smaller than the total number of candidate questions, the effective graph remains sparse, preventing additional overhead from becoming a computational bottleneck.

## 4. Theoretical Results

We additionally establish theoretical justification for the proposed approach. First, we prove that our two-stage selection algorithm achieves near-optimality under submodularity assumptions. Second, we provide justification for training objective in Eq. (1) with Fact 2.1 in our framework.

**Near-optimal guarantee for greedy selection.** The adaptive elicitation strategy in Section 3.2 requires solving a sequential decision problem: at each round, we select a pair $(R_t, x_t)$ (a subgroup of respondents and a query) to maximize the cumulative information gain over a horizon $T$. However, finding the optimal sequence $\{(R_t^*, x_t^*)\}_{t=1}^T$ is computationally intractable due to its combinatorial complexity over both the user set $\mathcal{V}$ and the query space $\mathcal{X}$ as shown in (Krause et al., 2008). We therefore adopt a two stage greedy algorithm in Section 3.2 that selects the pair $(R_t, x_t)$ at each step to maximize immediate information gain given the current interaction history. To justify this approach, we show that under standard submodularity and monotonicity assumptions and certain alignment assumption, both the joint and two stage greedy strategies achieve near optimal solutions, provably within a constant factor of the optimum.

Let $f : 2^{\mathcal{V} \times \mathcal{X}} \to \mathbb{R}_{\geq 0}$ be a utility function that measures the information gained about the group's latent properties, such as the expected information gain (EIG) defined in Section 3.2. The marginal gain of adding a new subset $A \subset \mathcal{V} \times \mathcal{X}$ to history $S$ is defined as $\Delta(A \mid S) = f(S \cup A) - f(S)$. We

now prove the following two theorems on near-optimality of greedy algorithms.

**Theorem 4.1** (Near optimality of joint greedy selection). *Under Assumption A.1, let $\{(R_t, x_t)\}_{t=1}^T$ be the sequence selected by the greedy algorithm, where at each step:*

$$(R_t, x_t) = \arg \max_{\substack{R \subset \mathcal{V}, |R| \le k \\ x \in \mathcal{X} \setminus \{x_1, \dots, x_{t-1}\}}} \Delta(R \times \{x\} \mid S_{t-1}), \quad (6)$$

*with $S_0 = \emptyset$ and $S_t = S_{t-1} \cup (R_t \times \{x_t\})$. If $\{(R_t^*, x_t^*)\}_{t=1}^T$ is the optimal sequence, then there exists a constant $C_1$ independent of $k$ and $T$ such that the inequality below holds:*

$$f\left(\cup_{t=1}^T (R_t^* \times \{x_t^*\})\right) \le C_1 \cdot f\left(\cup_{t=1}^T (R_t \times \{x_t\})\right).$$

**Theorem 4.2** (Near optimality of two-stage greedy algorithm). *Under Assumptions A.1, let $\{(R_t, x_t)\}_{t=1}^T$ be the sequence selected by the two-stage greedy algorithm below:*

$$\begin{aligned}
x_t &= \arg \max_{x \in \mathcal{X} \setminus \{x_1, \dots, x_{t-1}\}} \Delta((\mathcal{V} \times \{x\}) \mid S_{t-1}), \\
V_t &= \arg \max_{R \subset \mathcal{V}, |V| \le k} \Delta(R \times \{x_t\} \mid S_{t-1}),
\end{aligned} \quad (7)$$

*with $S_0 = \emptyset$ and $S_t = S_{t-1} \cup (R_t \times \{x_t\})$. If $\{(R_t^*, x_t^*)\}_{t=1}^T$ is the optimal sequence, then there exists a constant $C_2$ independent of $k$ and $T$ such that the inequality below holds:*

$$f\left(\cup_{t=1}^T (R_t^* \times \{x_t^*\})\right) \le C_2 \cdot f\left(\cup_{t=1}^T (R_t \times \{x_t\})_{t=1}^T\right).$$

The above results extend the classic guarantee for submodular utility maximization proved in (Krause et al., 2008; Wang et al., 2025) to our setting of joint user-query selection, ensuring near optimality of greedy algorithms.

*Remark* 4.3 (Greedy vs. Multi-step Planning). Theorem 4.1 and 4.2 guarantee that the greedy algorithm is already near optimal. This is further supported by our empirical results in Section 5.5, where more complex multi step planning methods provide only marginal improvements while incurring substantial computational cost. Although finite horizon planning may achieve higher information gain in principle, the greedy algorithm's combination of tractability and provable near optimality makes it especially well-suited for our principled adaptive group elicitation framework, where efficiency is essential for real-time interaction.

**De Finetti's theorem and meta-training.** For individual-level query–response processes where a latent entity $U$ governs a group member's behavior, as is the cases of political survey and student assessment, Remark A.6 proves the martingale condition holds and Theorem A.10 shows, conditional on the history $\mathcal{H}_T$, predictive inference can recover the latent entity $U$. Thus, the training objective of the LLM in Eq. (1) is designed to directly minimize the KL-divergence between the predictive distribution induced by LLM and the true data-generating process, ensuring unbiased belief updating and uncertainty quantification about latent entity $U$ similar as in (Wang et al., 2025).

## 5. Experiments

Our experimental evaluation is structured around the following questions:

- **RQ1:** Does adaptive group elicitation improve inference under different level of query budgets? → *Sec. 5.2*
- **RQ2:** What drives gains from group-relational respondent selection, and who benefits most? → *Sec. 5.3*
- **RQ3:** How do query selection and GNN imputation contribute to performance? → *Sec. 5.4*
- **RQ4:** How does multi-step planning compare to greedy selection in practice? → *Sec. 5.5*

### 5.1. Experimental Setup

**Datasets.** We evaluate adaptive elicitation on three opinion datasets: CES (Schaffner & Ansolabehere, 2025), a nationally representative U.S. election survey; OPINIONQA (Santurkar et al., 2023), a large-scale collection of public opinion questions across social and political topics; and TWIN-2K (Toubia et al., 2025), which measures economic preferences, cognitive biases, and personality traits for 2,000 individuals. All datasets consist of multiple-choice questions with demographic attributes, enabling controlled evaluation of population-level inference under constrained observation budgets. Additional details are provided in Appendix C.1.

**Evaluation Protocol and Metrics.** Methods iteratively select both *questions* and *respondents* over multiple rounds under a fixed observation budget. Performance is evaluated on a disjoint set of *target questions* never queried during elicitation, measuring the ability to infer unobserved responses. We report average predictive accuracy on target questions as the primary metric, with Perplexity (Jelinek et al., 1977) and Brier Score (Glenn et al., 1950) used to assess calibration (Appendix C.2). Unless stated otherwise, results are averaged over 10 independent trials per dataset, each with 20 candidate questions and 5 target questions.

**Models and Training.** We meta-train our query-selection policy by fine-tuning a pretrained Llama-3.1-8B model with LoRA and respondent-level data splits, using 80/20 train/validation in the in-distribution region (South) and holding out all respondents from the out-of-distribution region (West) for testing. We select the final checkpoint by validation loss, and additionally report results with a smaller Llama-3.2-1B model trained under the same protocol (full fine-tuning) in Appendix D.3. For population modeling and imputation, we train a heterogeneous R-GCN on a graph with respondent, demographic-feature, and query–choice nodes, with all splits performed at the user level to prevent leakage. The GNN is trained under simulated partial observability (including partial-edge masking and cold-start users), and evaluation is conducted in a strict cold-start setting where all user–query edges are removed from the

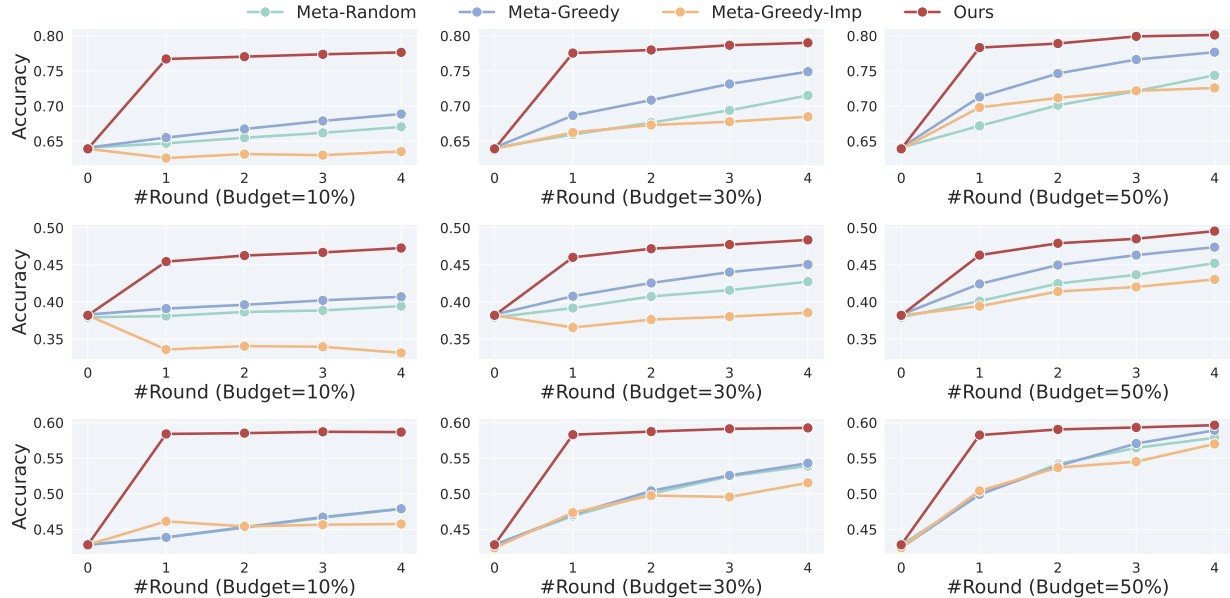

*Figure 3.* Accuracy across interaction rounds under different query budgets. **Upper**: CES. **Middle**: OPINIONQA. **Lower**: TWIN-2K.

message-passing graph. See Appendix C.4 for more details.

**Baseline Methods.**

We compare our method against three baselines that vary in their query selection strategy and imputation mechanism, including variants that incorporate ideas from prior work (Wang et al., 2025):

- *Meta-Random*. This baseline selects questions uniformly at random at each interaction round. A pretrained LLM is used to predict responses to held-out queries from the interaction history. No imputation is performed; responses to unqueried questions are neither imputed nor included in the interaction history.
- *Meta-Greedy*. Following the meta-policy framework of (Wang et al., 2025), this baseline uses a pretrained LLM to greedily select queries that maximize expected information gain given the current interaction history. At each round, the model scores all candidate questions based on their predicted informativeness and selects the highest-scoring one. No imputation is performed; responses to unqueried questions are neither imputed nor included in the interaction history.
- *Meta-Greedy-Imp*. This baseline extends *Meta-Greedy* by incorporating LLM-based imputation of missing responses. After each interaction round, a pretrained LLM predicts responses for respondents that have not been queried, conditioned on the observed interaction history. These predicted responses are treated as observed when selecting subsequent queries; however, imputation is performed independently for each respondent and does not exploit population-level structure.

Here, Meta denotes methods that adapt a pretrained LLM as a conditional query policy following the formulation in (Wang et al., 2025). Implementation details are deferred to Appendix C.4.

### 5.2. Overall Gains from Adaptive Elicitation

Figure 3 (a–c) reports accuracy on target questions across interaction rounds under different query budgets on three datasets. Across all datasets and budget levels, *Ours* consistently outperforms existing baselines. For example, at a 10% respondent budget on CES, our method achieves consistent gains over the strongest baseline, with relative improvements ranging from 17.1% at round 1 to 12.6% by round 4. These results indicate that effective adaptive elicitation requires jointly reasoning about *who to ask* and *what to ask*, rather than optimizing query selection alone. By explicitly modeling group structure and propagating partial observations, our framework allows information collected from a small subset of respondents to inform predictions for many others, leading to substantially more efficient use of limited budgets. Similar trends are observed for Perplexity and Brier Score (Appendix D.1). In addition, *Meta-Greedy-Imp* does not consistently outperform *Meta-Greedy*, suggesting that LLM-based imputation alone is unreliable without explicit population-level structure, and highlighting the importance of graph-based propagation.

### 5.3. When Does Respondent Selection Help Most?

We analyze our group-relational respondent selection to understand where adaptive elicitation gains arise and which re-

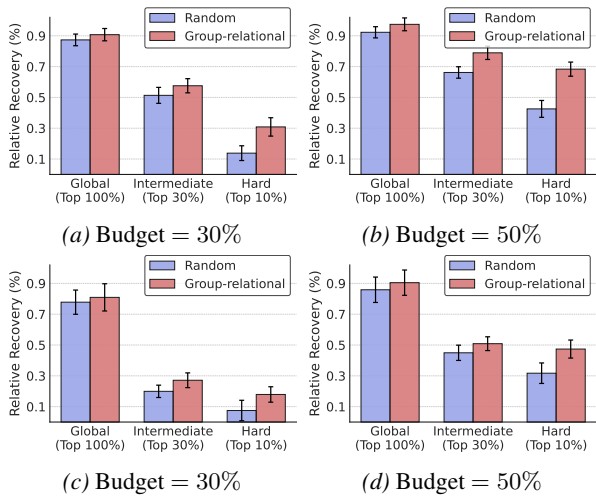

*(a)* Budget $= 30\%$  *(b)* Budget $= 50\%$

*(c)* Budget $= 30\%$  *(d)* Budget $= 50\%$

*Figure 4.* Relative recovery. Top $N\%$ denotes respondents in the highest $N\%$ by sensitivity. **(a)-(b)**: CES. **(c)-(d)**: OPINIONQA

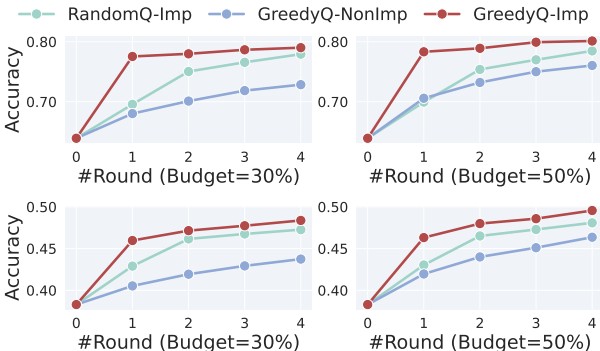

*Figure 5.* Ablation of query selection and imputation across datasets. **Upper**: CES. **Lower**: OPINIONQA.

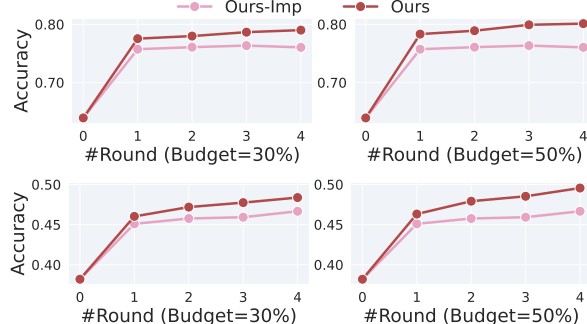

*Figure 6.* Imputation-only (*Ours-Imp*) vs. active observation (*Ours*) across rounds and budgets. **Upper**: CES. **Lower**: OPINIONQA.

spondents benefit most. While population-level propagation can amplify observed information, its effectiveness depends on which respondents are observed. We stratify respondents by *sensitivity*, defined as the potential gain from observing ground-truth responses, measured by the performance gap between full observation and imputation-only settings (see Appendix C.3). We report gains using *Relative Recovery*, $\text{RelRec}(t) = \frac{\text{Acc}(t) - \text{Acc}(0)}{\text{Acc}_{\text{Full}} - \text{Acc}(0)}$, which quantifies the fraction of the round-0 to full-observation gap recovered by round $t$. As shown in Figure 4, gains are highly non-uniform and concentrate on highly sensitive (hard) respondents, reaching up to 20% relative recovery under a 50% budget across both datasets. Absolute accuracy exhibits consistent trends across sensitivity tiers and budgets (see in Tables 2 and 3).

### 5.4. Ablations on Query Selection and Imputation

Figure 5 compares three variants that fix respondent selection while varying query selection (random vs. greedy) and imputation (with vs. without GNN-based propagation), isolating the contribution of each component across budgets and interaction rounds.

**Effect of Query Selection.** To isolate the effect of query selection, we compare *RandomQ-Imp* and *GreedyQ-Imp* while holding imputation fixed. Greedy query selection consistently achieves higher accuracy than random querying, with performance gaps widening over interaction rounds. These gains are stable across respondent budgets, indicating that selecting more informative queries yields reliable benefits once imputation is enabled.

**Effect of Imputation.** We compare *GreedyQ-NonImp* and *GreedyQ-Imp*, holding the query selection strategy fixed. *GreedyQ* selects questions greedily based on expected information gain, while *NonImp* and *Imp* respectively denote

disabling or enabling GNN-based imputation over the respondent graph. Enabling imputation yields substantially larger gains across all respondent budgets and interaction rounds. Without imputation, accuracy improves only modestly as more interactions are observed; in contrast, GNN-based imputation propagates partial observations across the group, enabling faster improvement and more sustained performance gains.

**Benefits of Actively Selecting Observations.** Figure 6 illustrates the effect of actively selecting which respondents to observe. *Ours-Imp* performs group-level imputation at each round without directly observing any respondent answers, instead leveraging population-level structure and group relations to infer responses. Comparing it with *Ours* shows that observing even a small number of strategically selected respondents per round yields consistent gains beyond imputation alone, highlighting the complementary roles of observation and graph-based propagation.

### 5.5. Multi-Step Planning: Are the Gains Worth It?

Table 1 compares greedy selection with multi-step planning across budgets and sensitivity tiers. Under multi-step planning, we first identifies the top-$k$=10 candidate queries according to the greedy criterion, then simulates their future impact by performing $N$=3 simulated rollouts per candidate, with each rollout unrolling up to the remaining inter-

*Table 1.* Round 4 accuracy (averaged over 5 trials) for greedy vs. multi-step planning under different budgets.

| Budget | Method | Global (100%) | Broad (50%) | Inter. (30%) | Hard (10%) | Extreme (5%) |
|---|---|---|---|---|---|---|
| 10% | Greedy | **0.488** | 0.426 | 0.384 | 0.356 | 0.322 |
| | Multi-step | 0.485 | **0.439** | **0.398** | **0.372** | **0.336** |
| 30% | Greedy | **0.500** | 0.464 | 0.438 | 0.441 | 0.432 |
| | Multi-step | 0.499 | **0.478** | **0.456** | **0.464** | **0.443** |
| 50% | Greedy | 0.507 | **0.510** | **0.507** | 0.545 | **0.561** |
| | Multi-step | 0.507 | 0.504 | 0.500 | **0.547** | 0.542 |

action steps. Rollouts approximate the downstream effect of a query by alternately selecting subsequent queries and imputing responses under the current belief state, and the candidate with the highest average rollout utility is selected. Consistent with Remark 4.3, this procedure yields at most marginal gains over greedy selection: performance differences on the global population are negligible, and improvements on highly sensitive subsets are small, inconsistent, and largely disappear or reverse at higher budgets (50%). Overall, these limited and unstable gains do not justify the substantially higher computational cost of multi-step planning.

## 6. Related Work

**Inference for graphical models** A number of previous works have focused on modeling relational structures within groups and analyzing data with inherent graphical properties, such as spatial outcomes. Early efforts, including Gelfand & Vounatsou (2003); Dobra et al. (2011); De Oliveira (2012), employed conditional autoregressive models (CAR) to model graphical signals, where a node's outcome is predicted using the outcomes of its neighboring nodes. While these models are interpretable and effective for spatial economics and biomedical data, they are parametric and struggle to handle complex natural language information which is a key component of tasks like political surveys, where queries and responses are often phrased in natural language. More recently, (Suh et al., 2025) proposed using heterogeneous GNNs to process group information without explicitly building a relational structure inside group. We adopt this approach and use the heterogeneous GNN as a tool for message propagation.

**Predictive Inference** De Finetti's predictive perspective on inference establishes that uncertainty originates from unobserved data. Berti et al. (2004; 2013); Fong et al. (2024) extend this viewpoint and propose computational schemes for predictive Bayesian inference. Building on the interpretation of In-Context Learning (ICL) as Bayesian inference (Xie et al., 2021; Ye & Namkoong, 2024), in (Wang et al., 2025), the authors demonstrate that fine-tuned large language models (LLMs) constitute valid tools for predictive inference. Our work adopts this approach by utilizing the ICL ability of LLMs as an effective predictive inference mechanism. This perspective naturally connects to con-

formal risk control (Chen et al., 2025; Snell et al., 2022; Deng et al., 2023; Zollo et al., 2024; Deng et al., 2025). In particular, while the predictive Bayesian view supplies a mechanism for quantifying uncertainty about unobserved outcomes, conformal risk control calibrates this uncertainty using held-out exchangeable data so that the resulting prediction sets, abstention rules, or selective decisions satisfy a prescribed risk level.

**Multi-turn Elicitation** Latent entity elicitation involves gathering information about unobservable characteristics, such as student ability, political intentions, or patient health status. Sequential Bayesian experimental design has been a predominant approach for such problems. Rainforth et al. (2024) provides a comprehensive review of modern Bayesian experimental design, ranging from Bayesian adaptive methods in (Cheng & Shen, 2005) to deep learning techniques adopted in (Foster et al., 2021; Ivanova et al., 2021). Recently, Wang et al. (2025) proposed using autoregressive models like LLMs without explicit prior specification. However, these methods are confined to individual-level inference. Our principle-based framework extends elicitation to group settings by incorporating the community's relational information, enabling uncertainty quantification for collective latent entities and group-aware selection of queries and respondents, a capability absent in prior work.

**Group interaction with LLM** Many studies (Hong et al., 2024; Khan et al., 2024; Tang et al., 2024; Luo et al., 2025; Du et al., 2026) study interactions among LLM agents, multi-agent systems, and social groups. However, collaborative decision-making frameworks do not directly fit our setting, which requires decentralized predictions for individual agents or group members from partial observations. To address this, we use the heterogeneous GNN approach of Suh et al. (2025) to impute unobserved responses and improve LLM-based evaluation, leveraging the language and reasoning capabilities of LLMs.

## 7. Conclusion

In this paper, we study adaptive group elicitation under limited query and respondent budgets, showing that effective elicitation requires jointly choosing questions, respondents, and mechanisms for propagating partial information. We propose a framework that combines adaptive query selection, group-relational respondent selection, and graph-based response imputation. Across real-world opinion datasets, our method improves accuracy and calibration under tight budgets, with especially strong gains in early rounds. These gains are most pronounced for highly sensitive respondents and remain robust across model scales, training regimes, and regions, suggesting that they stem from the elicitation and propagation mechanism rather than any specific backbone or population.

## Impact Statement

This paper presents work whose goal is to advance the field of machine learning. There are many potential societal consequences of our work, none of which we feel must be specifically highlighted here.

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

# A. Proof of Theoretical Results

In this section, we provide proofs and justification for results in Section 4 with details.

## A.1. Near Optimality of Greedy Algorithm

Recall the settings in Section 4, we let $S_t = \cup_{i=1}^t (R_i, x_i) \subset \mathcal{V} \times \mathcal{X}$ denote the set of user-query pairs selected up to round $t$, where by a slight abuse of notation, we shall always identify a pair $(R, x)$ of respondent subset $R \subset \mathcal{V}$ and query $x \in \mathcal{X}$ with the corresponding subset $R \times \{x\} \subset \mathcal{V} \times \mathcal{X}$. Let $f : 2^{\mathcal{V} \times \mathcal{X}} \to \mathbb{R}_{\geq 0}$ be a utility function that measures the information gained about the group's latent properties, such as the expected information gain defined in Section 3.2. The marginal gain of adding a new set of pairs $A \subset \mathcal{V} \times \mathcal{X}$ to a history $S$ is defined as $\Delta(A \mid S) = f(S \cup A) - f(S)$. For a set $S \subset \mathcal{V} \times \mathcal{X}$, we define the candidate query set $\mathcal{X}_S$ to be the set $\mathcal{X}_S = \{x \in \mathcal{X} : (\mathcal{V} \times \{x\}) \cap S = \emptyset\}$ consisting of queries that has not been selected. For fixed $k$, define $\Phi_k(x|S) = \max_{|R| \leq k} \Delta((R, x)|S)$, $\Psi(x|S) = \Delta((V, x)|S)$, and set $x_S^\Psi = \arg\max_{x \in \mathcal{X}_S} \Psi(x|S)$, the query with maximal utility gain when gather information from all respondents. Throughout this section, we assume $T > 2$, and we optimize over feasible $T$-round sequences $\{(R_t, x_t)\}_{t=1}^T$ with satisfying $|R_t| \leq k$ and that $x_1, \ldots, x_T$ are distinct, equivalently, at each step $t$, $x_t \in \mathcal{X}_{t-1}$.

**Assumption A.1.** The utility function $f$ satisfies the following properties for any subsets $W_1 \subseteq W_2 \subseteq \mathcal{V} \times \mathcal{X}$ and any $W \subset \mathcal{V} \times \mathcal{X}$:

(a) *(Zero-utility for empty set.)* $f(\emptyset) = 0$

(b) *(Monotonicity).* $f(W_2) \geq f(W_1)$.

(c) *(Submodularity).* $\Delta(W \mid W_1) \geq \Delta(W \mid W_2)$.

(d) *(Query-budget alignment).* There exists a constant $\eta \in (0, 1)$, such that for each $k$ and any set $S \subset \mathcal{V} \times \mathcal{X}$ with nonempty candidate set and $|\mathcal{X}_S| \geq 1$, we have

$$\Phi_k(x_S^\Psi|S) \geq \eta \max_{x \in \mathcal{X}_S} \Phi_k(x|S).$$

Monotonicity means that adding more observations does not decrease utility. Submodularity captures the principle of diminishing returns: the marginal benefit of adding a new observation diminishes as the history grows, these property is commonly assumed in adaptive information acquisition problems (Krause et al., 2008; Wang et al., 2025). The last alignment condition rules out candidate queries whose EIG is concentrated on an arbitrarily small niche subset of respondents. This is consistent with our setting, where candidate queries are designed to elicit information about a shared group-level latent entity instead of few specific individuals. Specifically, suppose there are constants $0 < \lambda^- \leq \lambda^+ < \infty$ and nontrivial gain $\Psi(x_S^\Psi) > 0$, such that for each budget $k \in [1 : |\mathcal{V}|]$, $\lambda^- \leq \frac{\Phi_k(x|S)}{\Psi(x|S)} \leq \lambda^+$ holds for any $x \in \mathcal{X}_S$, then by the following inequality,

$$\Phi_k(x_S^\Psi|S) \geq \lambda^- \Psi(x_S^\Psi|S) = \lambda^- \max_{x \in \mathcal{X}_S} \Psi(x|S) \geq \frac{\lambda^-}{\lambda^+} \max_{x \in \mathcal{X}_S} \Phi_k(x|S),$$

Assumption A.1(d) holds with $\eta = \lambda^-/\lambda^+$ uniformly for any set $S$ and budget $k$.

**Theorem A.2** (Restatement of Theorem 4.1). *Let $\{(R_t, x_t)\}_{t=1}^T$ be the sequence selected by the following joint greedy algorithm, where at each step $t$:*

$$x_t = \arg\max_{x \in \mathcal{X}_{S_{t-1}}} \Phi_k(x|S_{t-1}), \; R_t = \arg\max_{|R| \leq k, R \subset \mathcal{V}} \Delta((R, x_t)|S_{t-1}).$$

*with $S_0 = \emptyset$ and $S_t = S_{t-1} \cup (R_t, x_t)$. If $\{(R_t^*, x_t^*)\}_{t=1}^T$ is the optimal sequence, then under Assumption A.1:*

$$f(\cup_{t=1}^T (R_t^*, x_t^*)) \leq \frac{2}{1 - e^{-2}} \cdot f(\cup_{t=1}^T (R_t, x_t)).$$

*Proof of Theorem 4.1.* Assume that $f$ is monotone and submodular on the set of all pairs $(R, x)$ where $V \subset \mathcal{V}$ and $x \in \mathcal{X}_{\text{pool}}$, as per Assumption A.1. We aim to show that for the greedy sequence $\{(R_t, x_t)\}_{t=1}^T$ and the optimal sequence $\{(R_t^*, x_t^*)\}_{t=1}^T$, the following approximation holds:

$$f(S_T^*) \le \frac{2}{1 - e^{-2}} f(S_T)$$

where $S_i = \cup_{l=1}^i (R_l, x_l)$ for $i \ge 1$ and $S_0 = \emptyset$, and similarly for $S_T^*$. We assume $f(\emptyset) = 0$.

By monotonicity of $f$, for any $i$ with $0 \le i \le T$, $f(S_T^*) \le f(S_T^* \cup S_i)$. Since $S_i \subseteq S_T^* \cup S_i$, we can write:

$$f(S_T^* \cup S_i) = f(S_i) + \sum_{j=1}^T \left[ f(S_j^* \cup S_i) - f(S_{j-1}^* \cup S_i) \right]$$

where $S_j^*$ is the set of the first $j$ pairs in the optimal sequence. Fix $i$, denote $\Delta_j = f(S_j^* \cup S_i) - f(S_{j-1}^* \cup S_i)$, we have

$$f(S_T^*) \le f(S_i) + \sum_{j=1}^T \Delta_j.$$

Now we bound the terms $\Delta_j$ for $1 \le j \le T$. Let $X_i = \{x_1, \ldots, x_i\}$ be the set of queries in $S_i$. For each $j$, define:

- $\Gamma_i = \{j \in \{1, \ldots, T\} : x_j^* \in X_i\}$, the set of indices where the optimal query $x_j^*$ is in $S_i$

- For $j \in \Gamma_i$, let $\sigma_i(j)$ be the unique index in $\{1, \ldots, i\}$ such that $x_j^* = x_{\sigma_i(j)}$ (i.e., the greedy step where $x_j^*$ was selected)

We bound $\Delta_j$ in the following two cases.

**Case 1:** $j \in \Gamma_i$**.** Here, $x_j^*$ is already in $S_i$. By submodularity, since $S_{\sigma_i(j)-1} \subseteq S_i \subseteq S_{j-1}^* \cup S_i$:

$$\Delta_j \le f(S_{\sigma_i(j)-1} \cup (R_j^*, x_j^*)) - f(S_{\sigma_i(j)-1})$$

By the greedy choice at step $\sigma_i(j)$, the pair $(R_{\sigma_i(j)}, x_j^*)$ was chosen to maximize gain from $S_{\sigma_i(j)-1}$, thus

$$\Delta_j \le f(S_{\sigma_i(j)}) - f(S_{\sigma_i(j)-1}).$$

**Case 2:** $j \notin \Gamma_i$ Here, $x_j^*$ is not in $S_i$. By submodularity, since $S_i \subseteq S_{j-1}^* \cup S_i$:

$$\Delta_j \le f(S_i \cup (R_j^*, x_j^*)) - f(S_i)$$

At step $i + 1$, the greedy algorithm selects a pair $(R_{i+1}, x_{i+1})$ that maximizes the gain from $S_i$. Since $x_j^* \notin X_i$, the pair $(R_j^*, x_j^*)$ is considered at step $i + 1$, so:

$$f(S_i \cup (R_j^*, x_j^*)) - f(S_i) \le f(S_{i+1}) - f(S_i)$$

Thus we have $\Delta_j \le f(S_{i+1}) - f(S_i)$ in this case.

Splitting the sum over $j$ by the two different cases

$$\sum_{j=1}^T \Delta_j = \sum_{j \in \Gamma_i} \Delta_j + \sum_{j \notin \Gamma_i} \Delta_j,$$

from the above discussion, we have

$$\sum_{j \in \Gamma_i} \Delta_j \le \sum_{j \in \Gamma_i} \left[ f(S_{\sigma_i(j)}) - f(S_{\sigma_i(j)-1}) \right], \quad \sum_{j \notin \Gamma_i} \Delta_j \le \sum_{j \notin \Gamma_i} \left[ f(S_{i+1}) - f(S_i) \right]$$

Since $\sigma_i$ is an injection from $\Gamma_i$ to $[i]$, the sum over $j \in \Gamma_i$ covers distinct increments, by monotonicity, it holds that

$$\sum_{j \in \Gamma_i} \left[ f(S_{\sigma_i(j)}) - f(S_{\sigma_i(j)-1}) \right] \le \sum_{t=1}^i [f(S_t) - f(S_{t-1})] = f(S_i),$$

Note the number of $j \notin \Gamma_i$ is $T - |\Gamma_i| \leq T$, by monotonicity,

$$\sum_{j \notin \Gamma_i} [f(S_{i+1}) - f(S_i)] \leq (T - |\Gamma_i|)[f(S_{i+1}) - f(S_i)] \leq T[f(S_{i+1}) - f(S_i)].$$

Hence $\sum_{j=1}^{T} \Delta_j \leq f(S_i) + T[f(S_{i+1}) - f(S_i)]$. Substituting into the inequality for $f(S_T^*)$, we have

$$f(S_T^*) \leq 2f(S_i) + T[f(S_{i+1}) - f(S_i)].$$

Define $\delta_i = f(S_T^*) - 2f(S_i)$. Then by the above inequality,

$$\delta_{i+1} \leq \delta_i - 2 \cdot \frac{\delta_i}{T} = \delta_i \left(1 - \frac{2}{T}\right)$$

Note that $\delta_0 = f(S_T^*)$ since $f(S_0) = 0$, we have:

$$\delta_T \leq \delta_0 \left(1 - \frac{2}{T}\right)^T \leq f(S_T^*)e^{-2}$$

Rearranging gives $f(S_T^*) \leq \frac{2}{1-e^{-2}} f(S_T)$. $\qquad\square$

**Theorem A.3** (Near optimality of two-stage greedy algorithm). *Let $\{(R_t, x_t)\}_{t=1}^T$ be the sequence selected by the two-stage greedy algorithm, where at each step:*

$$x_{t+1} = \arg \max_{x \in \mathcal{X}_{S_t}} \Delta((\mathcal{V}, x) \mid S_t), \quad R_{t+1} = \arg \max_{R \subset \mathcal{V}, |R| \leq k} \Delta((R, x_{t+1}) \mid S_t),$$

*with $S_0 = \emptyset$ and $S_t = S_{t-1} \cup (R_t, x_t)$. If $\{(R_t^*, x_t^*)\}_{t=1}^T$ is the optimal sequence, then under Assumptions A.1:*

$$f\left(\cup_{t=1}^T (R_t^*, x_t^*)\right) \leq \frac{2}{1 - e^{-2\eta}} \cdot f\left(\cup_{t=1}^T (R_t, x_t)\right).$$

*Proof of Theorem A.3.* Assume that $f$ is monotone and submodular on the set of all pairs $(R, x)$ where $R \subset \mathcal{V}$ and $x \in \mathcal{X}_{\text{pool}}$. We aim to show that for the two-stage greedy sequence $\{(R_t, x_t)\}_{t=1}^T$ and the optimal sequence $\{(R_t^*, x_t^*)\}_{t=1}^T$, the following approximation holds:

$$f(S_T^*) \leq \frac{2}{1 - e^{-2\eta}} f(S_T)$$

where $S_i = \cup_{l=1}^i (R_l, x_l)$ for $i \geq 1$ and $S_0 = \emptyset$, and similarly for $S_T^*$. By Assumption A.1, we have $f(S_0) = f(\emptyset) = 0$. By monotonicity of $f$, for any $i$ with $0 \leq i \leq T$, $f(S_T^*) \leq f(S_T^* \cup S_i)$. Since $S_i \subseteq S_T^* \cup S_i$, we can write:

$$f(S_T^* \cup S_i) = f(S_i) + \sum_{j=1}^T \left[f(S_j^* \cup S_i) - f(S_{j-1}^* \cup S_i)\right]$$

where $S_j^*$ is the set of the first $j$ pairs in the optimal sequence. Fix $i$, denote $\Delta_j = f(S_j^* \cup S_i) - f(S_{j-1}^* \cup S_i)$, we have

$$f(S_T^*) \leq f(S_i) + \sum_{j=1}^T \Delta_j.$$

Now we bound the terms $\Delta_j$ for $1 \leq j \leq T$. Let $X_i = \{x_1, \ldots, x_i\}$ be the set of queries in $S_i$. For each $j$, define:

- $\Gamma_i = \{j \in \{1, \ldots, T\} : x_j^* \in X_i\}$, the set of indices where the optimal query $x_j^*$ is in $S_i$

- For $j \in \Gamma_i$, let $\sigma_i(j)$ be the unique index in $\{1, \ldots, i\}$ such that $x_j^* = x_{\sigma_i(j)}$ (i.e., the greedy step where $x_j^*$ was selected)

We bound $\Delta_j$ in the following two cases.

**Case 1:** $j \in \Gamma_i$. Here, $x_j^*$ is already in $S_i$. By submodularity, since $S_{\sigma_i(j)-1} \subseteq S_i \subseteq S_{j-1}^* \cup S_i$:

$$\Delta_j = \Delta\big((R_j^*, x_j^*)|S_{\sigma_{j-1}^*} \cup S_i\big) \leq \Delta\big((R_j^*, x_j^*)|S_{\sigma_i(j)-1}\big).$$

By the greedy choice at step $\sigma_i(j)$, we have

$$\Delta\big((R_j^*, x_j^*)|S_{\sigma_i(j)-1}\big) \leq \Delta\big((R_j^*, x_j^*)|S_{\sigma_i(j)-1}\big)$$

thus

$$\Delta_j \leq \Delta\big((R_j^*, x_j^*)|S_{\sigma_i(j)-1}\big) \leq \Delta\big((R_j^*, x_j^*)|S_{\sigma_i(j)-1}\big) = f(S_{\sigma_i(j)}) - f(S_{\sigma_i(j)-1}).$$

**Case 2:** $j \notin \Gamma_i$ Here, $x_j^*$ is not in $S_i$. By submodularity, since $S_i \subseteq S_{j-1}^* \cup S_i$:

$$\Delta_j \leq f(S_i \cup (R_j^*, x_j^*)) - f(S_i)$$

By Assumption A.1(d) and the two-stage greedy selection: we have $x_{S_i}^\Psi = x_{i+1}$, and $\Phi_k(x_{i+1}|S_i) = f(S_{i+1}) - f(S_i)$, and

$$\Delta\big((R_j^*, x_j^*)|S_i\big) \leq \max_{x \in \mathcal{X}_{S_i}} \Phi_k(x|S_i) \leq \frac{1}{\eta} \Phi_k(x_S^\Psi|S) = \frac{1}{\eta}\big[f(S_{i+1}) - f(S_i)\big],$$

thus

$$\Delta_j \leq \frac{1}{\eta}\big[f(S_{i+1}) - f(S_i)\big]$$

Splitting the sum over $j$ by the two different cases

$$\sum_{j=1}^T \Delta_j = \sum_{j \in \Gamma_i} \Delta_j + \sum_{j \notin \Gamma_i} \Delta_j,$$

from the above discussion, we have

$$\sum_{j \in \Gamma_i} \Delta_j \leq \sum_{j \in \Gamma_i} \big[f(S_{\sigma_i(j)}) - f(S_{\sigma_i(j)-1})\big]$$

$$\sum_{j \notin \Gamma_i} \Delta_j \leq \frac{T - |\Gamma_i|}{\eta}\big[f(S_{i+1}) - f(S_i)\big] \leq \frac{T}{\eta}\big[f(S_{i+1}) - f(S_i)\big]$$

Since $\sigma_i$ is an injection from $\Gamma_i$ to $[i]$, the sum over $j \in \Gamma_i$ covers distinct increments, by monotonicity:

$$\sum_{j \in \Gamma_i} \big[f(S_{\sigma_i(j)}) - f(S_{\sigma_i(j)-1})\big] \leq \sum_{t=1}^i [f(S_t) - f(S_{t-1})] = f(S_i)$$

Hence $\sum_{j=1}^T \Delta_j \leq f(S_i) + \frac{T}{\eta}[f(S_{i+1}) - f(S_i)]$. Substituting into the inequality for $f(S_T^*)$, we have

$$f(S_T^*) \leq 2f(S_i) + \frac{T}{\eta}[f(S_{i+1}) - f(S_i)]$$

Define $\delta_i = f(S_T^*) - 2f(S_i)$. Then by the above inequality,

$$\delta_{i+1} = \delta_i - 2[f(S_{i+1}) - f(S_i)] \leq \delta_i - 2 \cdot \frac{\eta}{T}\delta_i = \delta_i\left(1 - \frac{2\eta}{T}\right)$$

Note that $\delta_0 = f(S_T^*)$ since $f(S_0) = 0$, we have:

$$\delta_T \leq \delta_0\left(1 - \frac{2\eta}{T}\right)^T \leq f(S_T^*)e^{-2\eta}$$

Rearranging gives $f(S_T^*) \leq \frac{2}{1 - e^{-2\eta}}f(S_T)$. $\qquad \square$

## A.2. Generalization of de Finetti Theorem and Related Justification

This section presents the theoretical foundation for Fact 2.1, which generalizes de Finetti's theorem, and establishes the martingale condition Remark A.6 and frequentist consistency Theorem A.10 of the copula-based predictive inference framework. This justification ensures that our method provides valid uncertainty quantification under the martingale condition, extending Bayesian principles to sequential prediction settings without explicit likelihood specification (Ye & Namkoong, 2024; Wang et al., 2025).

We begin by introducing key definitions. The conditionally identically distributed (c.i.d.) sequence serves as the foundational concept for predictive inference, generalizing exchangeable sequences through a martingale formulation that accommodates broader data-generating processes. In the following context, we shall focus on the real random variable case exclusively, and we remark here that (Berti et al., 2004) established Theorem A.7 and limit theorems for random variables taking value in general Polish space. Specifically, we fix a probability space $(\Omega, \mathbb{P}, \mathcal{F})$ and consider a random variable sequence $\{Y_i\}_{i=1}^{\infty}$ taking value in $\mathbb{R}$. For $y \in \mathbb{R}$, denote $P_i(y) = \mathbb{P}(Y_{i+1} \leq y | Y_{1:i})$. Throughout this subsection, we shall always use capital letter $P$ represent cumulative distribution function, and small letter $p$ represent distribution density function.

**Definition A.4** (Conditionally identically distributed (c.i.d.) sequence). A sequence of random variables $Y_1, Y_2, \ldots$ is conditionally identically distributed (c.i.d.) if, for all $i \geq 1$ and all $k \geq 1$, the following holds almost surely:

$$\mathbb{P}(Y_{i+k} \leq y \mid Y_1, \ldots, Y_i) = P_i(y)$$

where $P_i(\cdot)$ denotes the predictive cumulative distribution function of $Y_{i+1}$ after observing $Y_1, \ldots, Y_i$. This implies that, conditional on the past, all future observations are identically distributed according to the current predictive distribution.

The following martingale formulation of c.i.d. sequences provides the theoretical basis for the approaches in (Fong et al., 2024; Ye & Namkoong, 2024) and our work. This formulation is particularly valuable as it enables the application of martingale convergence tools to predictive inference.

**Lemma A.5** (Martingale formulation, (Berti et al., 2004)). *For a sequence of random variables $Y_{1:\infty}$, it is conditionally identically distributed (c.i.d.) if and only if the sequence of predictive distribution functions $\{P_i(y)\}_{i=1}^{\infty}$ constitutes a bounded martingale for each $y$. That is, for $i > 1$ and $y \in \mathbb{R}$, the following equality holds*

$$\mathbb{E}[P_i(y) \mid Y_1, \ldots, Y_{i-1}] = P_{i-1}(y). \tag{8}$$

*This martingale property ensures unbiased updating of the predictive distribution sequence, a condition termed predictive coherence in (Fong et al., 2024).*

Now we first provide a basic results that justifies the martingale assumption in the individual-level elicitation setting.

*Remark* A.6. Let $Y_1, Y_2, \ldots$ be an exchangeable sequence of random variables. Under the Bayesian model that first a latent entity $U$ is i.i.d drawn from a prior $\mu(U)$, and then for this fixed $U$, $\{Y_i\}_{i=1}^{\infty}$ are i.i.d drawn from the distribution $\mathbb{P}(\cdot|U)$, the sequence of predictive distributions $M_t(y) = \mathbb{P}(Y_{t+1} \leq y \mid Y_{1:t})$ forms a martingale with respect to the filtration $\mathcal{F}_t = \sigma(Y_{1:t})$ generated by $Y_{1:t}$. That is, for all $t \geq 1$:

$$\mathbb{E}[M_{t+1} \mid \mathcal{F}_t] = M_t \quad \text{almost surely.}$$

This martingale property justifies our assumption in Section 4 that the predictive distributions of underlying data generation process satisfy the martingale condition, as it demonstrates that whenever there exists a latent entity $U$ governing the observed outcomes, for example, when $Y_1, Y_2, \ldots$ are conditionally i.i.d given $U$. This covers the cases including political surveys, student assessments, and other group elicitation scenarios. the sequence of predictive beliefs naturally exhibits martingale behavior under coherent Bayesian updating.

*Proof of Remark A.6.* By the law of total expectation and Bayesian updating:

$$\mathbb{E}[M_{t+1}(y) \mid Y_{1:t}] = \mathbb{E}[\mathbb{P}(Y_{t+2} \leq y \mid Y_{1:t+1}) \mid Y_{1:t}]$$
$$= \int \mathbb{P}(Y_{t+2} \leq y \mid Y_{1:t+1})\mathbb{P}(Y_{t+1} \mid Y_{1:t})dY_{t+1}.$$

Substituting the Bayesian predictive distributions gives: for each $y \in \mathbb{R}$,

$$\mathbb{E}[M_{t+1}(y) \mid Y_{1:t}] = \int \left( \int \mathbb{P}(Y_{t+2} \leq y \mid U) d\mu(U \mid Y_{1:t+1}) \right) \mathbb{P}(Y_{t+1} \mid Y_{1:t}) dY_{t+1}.$$

Using Bayes' rule for the posterior update, we have:

$$\mathbb{E}[M_{t+1}(y) \mid Y_{1:t}] = \int \int \mathbb{P}(Y_{t+2} \leq y \mid U) \frac{\mathbb{P}(y_{t+1} \mid U) d\mu(U \mid Y_{1:t})}{\mathbb{P}(y_{t+1} \mid Y_{1:t})} \mathbb{P}(y_{t+1} \mid Y_{1:t}) dy_{t+1}$$

$$= \int \int \mathbb{P}(Y_{t+2} \leq y \mid U) \mathbb{P}(y_{t+1} \mid U) d\mu(U \mid Y_{1:t}) dy_{t+1}.$$

By Fubini's theorem and the fact that $\int \mathbb{P}(Y_{t+1} \mid U) dY_{t+1} = 1$, we obtain $\mathbb{P}(Y_{t+2} \leq y \mid Y_{1:t}) = p(Y_{t+1} \leq y \mid Y_{1:t}) = M_t(y)$, completing the proof. $\qquad\square$

The martingale property is crucial as it facilitates the application of convergence theorems. The following theorem justifies the existence of a limiting distribution under the martingale condition, which is called *martingale posterior* in (Fong et al., 2024), generalizing the traditional de Finetti theorem. While de Finetti's theorem guarantees a latent entity governing data generation under exchangeability, this extension accommodates the broader class of c.i.d. sequences, making it applicable to non-exchangeable settings commonly encountered in practical machine learning scenarios.

**Theorem A.7** ((Berti et al., 2004)). *For a c.i.d. sequence, the sequence of predictive distributions $P_N$ converges weakly to a limiting random probability measure $P_\infty$ almost surely as $N \to \infty$.*

As demonstrated in the following Theorem A.10, in settings where a latent entity governs data generation, this limiting distribution effectively recovers the distribution density $\mathbb{P}(\cdot|U)$ induced by the latent entity.

**Copula-based computational scheme** To operationalize this theoretical framework, (Fong et al., 2024) established that any predictive distribution sequence satisfying the martingale condition in Eq. (8) necessarily arises from a copula-based updating rule. This provides a constructive method for building valid predictive inference machines. We now introduce the bivariate copula concept before presenting the key lemma.

**Definition A.8** (Bivariate Copula). A bivariate copula $C : [0,1]^2 \to [0,1]$ is a cumulative distribution function on the unit square with uniform marginal distributions. Its density is denoted $c(u,v)$. Sklar's Theorem establishes that any bivariate distribution can be expressed through its marginals and a copula capturing the dependence structure.

In the copula construction, we assume that each predictive distribution admits a density $p_i$ and a continuous CDF $P_i$.

**Lemma A.9** (Corollary 1 in (Fong et al., 2024)). *The sequence of conditional densities $p_0, p_1, \ldots$ satisfies the martingale condition Eq. (8) if and only if there exists a unique sequence of bivariate copula densities $c_1, c_2, \ldots$ such that*

$$p_{i+1}(y) = c_{i+1}\{P_i(y), P_i(y_{i+1})\} p_i(y)$$

*for $i \in \{0, 1, \ldots\}$, where $P_i$ represents the cumulative distribution function corresponding to $p_i$.*

Lemma A.9 guarantees that the following copula-based update rule satisfies the martingale condition in Eq. (8). The Gaussian copula provides a practical choice due to its parametric simplicity and flexibility in modeling dependencies. Specifically, let $\{\alpha_n\}_{n=1}^{\infty}$ be a deterministic learning rate sequence with $\alpha_n = \mathcal{O}(1/n)$, and let $c_\rho$ denote the density of the Gaussian copula with correlation parameter $\rho$. Denoting by $P_i(\cdot)$ the cumulative distribution function corresponding to $p_i(\cdot)$, the following update rule after observing $y_{i+1}$ induces a predictive distribution sequence satisfying the martingale condition:

$$p_{i+1}(y) = [1 - \alpha_{i+1} + \alpha_{i+1} c_\rho\{P_i(y), P_i(y_{i+1})\}] p_i(y). \qquad (9)$$

The following Theorem A.10 establishes the frequentist consistency of the predictive density $p_n$ estimated via the copula method. It demonstrates that $p_n$ converges to the true data-generating density $f_0$ when data $Y_{1:n} \overset{iid}{\sim} f_0$. This consistency result validates predictive inference under the martingale condition, ensuring that we can recover the latent entity from the limiting distribution. The combination of Theorems A.7 and A.10 guarantees that our predictive inference framework

provides a valid approach for latent entity recovery and uncertainty quantification through predictive sampling. Specifically, Theorem A.7 generalizes de Finetti's representation by ensuring the existence of a limiting martingale posterior for c.i.d. sequences, while Theorem A.10 ensures that this posterior converges to the true data-generating distribution, thereby if this response law is the distribution induced by a latent entity $U$, then the method recovers $U$ up to its induced predictive distribution $\mathbb{P}(\cdot|U)$.

**Theorem A.10.** *Assume $Y_1, Y_2, \ldots \sim f_0$ where $f_0$ is a density function on $\mathbb{R}$, then under the following conditions:*

*(a) $\rho \in (0,1)$ and the learning rate sequence is $\alpha_i = a(i+1)^{-1}$ with $a < 2/5$,*

*(b) There exists $B < \infty$ such that $f_0(y)/p_0(y) \leq B$ for all $y \in \mathbb{R}$ (i.e., the initial $p_0$ has tails at least as heavy as $f_0$),*

*the sequence $p_n$ given by the updating rule Eq. (9) is Hellinger consistent at $f_0$, that is,*

$$\lim_{n \to \infty} H^2(p_n, f_0) = 0 \quad \text{almost surely,}$$

*where the squared Hellinger distance is defined as $H^2(p, f) = 1 - \int \sqrt{p(y)f(y)}dy$.*

For the i.i.d. data generation process with ground truth density $f_0$, the distribution induced by the true latent entity is identified with the distribution with density $f_0$. Theorem A.10 shows that the copula predictive density $p_n$ converges to $f_0$ in Hellinger distance almost surely. Therefore, in this setting, the predictive update recovers the latent entity $U$ at the level of its induced distribution.

## B. Testing-Time Inference Details

In this section, we describe details about our methodology presented in Section 3.2. Recall that at test time, we are given a new group of members $\mathcal{V}^{\text{test}}$, a candidate query set $\mathcal{X}_c \subset \mathcal{X}$, and a held-out evaluation set $\mathcal{X}_h \subset \mathcal{X}\backslash\mathcal{X}_c$, and we have construct the heterogeneous graph for the testing group $\mathcal{V}^{test}$ following the same procedure as in the training phase with the same set of query-choice nodes and feature nodes as defined during training.

At interaction round $t$, suppose we have already selected query $X_t$ and subgroup of respondents $R_t \subset \mathcal{V}$. After collecting actual responses $Y^{R_t}$ for the selected query $X_t$ from the chosen subgroup of respondents $R_t$, we add new edges between group member nodes and query-choice nodes corresponding to the observation in the heterogeneous graph. We then perform message passing using the pre-trained GNN over the updated graph structure to compute refined node embeddings. These updated embeddings enable more accurate imputation of unobserved user responses to query $X_t$. For a group member node $v \notin V_t$, the prediction of the unobserved response $Y_t^v$ is obtained using Eq. (2). The imputed predictions for unobserved responses are added into the interaction history to inform the next round of adaptive selection. Specifically, we set $\mathcal{H}_t = \mathcal{H}_{t-1} \cup \{(X_t, \hat{Y}_t^{\mathcal{V}^{\text{test}}})\}$, where $\hat{Y}_t^{\mathcal{V}^{\text{test}}_m}$ is the concatenation of observation $Y_t^{R_t}$ for the subgroup of queried members and the GNN imputation $Y_t^{\mathcal{V}-R_t}$ for unqueried members. Then we add the edges corresponding to $\hat{\mathcal{Y}}^{\mathcal{V}^{test}}$ into the heterogeneous graph and perform message passing to update node embeddings. This enrichment of the graph structure allows the GNN to propagate the newly acquired information throughout the network. Consequently, the integration of both collected and imputed responses effectively reduces uncertainty about the group's latent entity. This is achieved through two mechanisms: the updated node embeddings enhance the accuracy of future GNN-based imputations, while the more complete individual-level interaction history improves the LLM's uncertainty quantification for subsequent adaptive query selections.

## C. Experiment Details

### C.1. Datasets

We evaluate group adaptive elicitation on three real-world opinion datasets spanning political attitudes, social values, and economic preferences. All datasets consist of real human responses collected via probability-based or nationally representative survey panels. Below, we summarize each dataset with respect to its data source, preprocessing procedures, demographic attributes, question selection, and experimental usage.

**CES.** The *Cooperative Election Study* (CES)[1] is one of the largest academic surveys of the American public and a core resource for studying U.S. political attitudes and electoral behavior. We use the 2024 wave of the CES and filter out respondents with missing values in either demographic attributes or selected opinion questions, resulting in a final sample of 3,326 respondents. We retain eight demographic variables capturing age cohort, race, gender, education, family income, religion, party identification, and political ideology. From the survey, we extract 30 policy questions, each formulated as a binary stance indicating whether the respondent *supports* or *opposes* a given policy. The selected questions span major policy domains, including health care, gun control, immigration, abortion, climate policy, policing and public safety, and executive authority (e.g., executive orders). These questions are chosen to ensure topical diversity, substantial response variation, and a clear stance-based interpretation suitable for adaptive elicitation.

**OPINIONQA.** OPINIONQA is derived from the *American Trends Panel* (ATP)[2], Pew Research Center's primary probability-based panel for U.S. public opinion research, which consists of approximately 10,000 adults recruited to be nationally representative of the U.S. population, with surveys administered in English and Spanish. We use three ATP waves—W50, W54, and W92—covering topics related to American families, economic inequality, and political typology. After filtering respondents with missing demographic information or incomplete question responses, the resulting dataset contains 2,368 respondents. We include eight standardized demographic variables capturing age category, gender, education, party affiliation, political ideology, income, religion, and race/ethnicity. We extract 30 opinion questions, each with 3–5 discrete response options (e.g., degrees of agreement or categorical preferences), enabling evaluation of adaptive elicitation beyond binary support–oppose judgments.

**TWIN-2K.** TWIN-2K[3] is a four-wave, nationally representative U.S. panel fielded in January–February 2025 on Prolific, designed for studying LLM-based human simulation. Participants complete questions spanning demographics, personality scales, cognitive ability tests, economic preferences, and heuristics-and-biases experiments. We exclude respondents with missing demographic attributes or incomplete responses on selected questions, yielding a final sample of 2,368 respondents. Demographic variables include age, sex, education, political party, political ideology, income, religion, and race. From the full survey, we select 40 questions primarily related to economic preferences and behavioral decision-making, which are well-suited for evaluating whether adaptive elicitation can infer latent economic and behavioral traits from limited observations.

For all three datasets, we exploit geographic variation to construct a realistic generalization setting. Respondents from the *South* region are used for LLM pretraining and meta-learning, including learning priors over response distributions and query utilities, while respondents from the *West* region are used for adaptive elicitation inference, where the model actively selects queries and infers unobserved responses. Additional cross-region results are reported in Appendix D.4 to assess robustness and geographic transferability. This regional split mirrors practical deployment scenarios in which elicitation policies learned from one population must generalize to related but distributionally distinct groups.

### C.2. Evaluation Metrics

**Accuracy.** Let $p_i \in \mathbb{R}^C$ denote the predicted probability distribution over $C$ answer options for instance $i$, and let $y_i \in \{1, \dots, C\}$ be the ground-truth class index. Accuracy is defined as:

$$\text{Acc} = \frac{1}{N} \sum_{i=1}^{N} \mathbb{I}\left[\arg\max_c \ p_{i,c} = y_i\right], \tag{10}$$

where $N$ is the number of evaluated instances and $\mathbb{I}[\cdot]$ is the indicator function.

**Perplexity (PPL).** Perplexity is defined as the exponential of the average negative log-likelihood:

$$\text{PPL} = \exp\left(-\frac{1}{N} \sum_{i=1}^{N} \log p_i(y_i)\right), \tag{11}$$

---

[1] https://dataverse.harvard.edu/dataset.xhtml?persistentId=doi:10.7910/DVN/X11EP6
[2] https://www.pewresearch.org/american-trends-panel-datasets/
[3] https://huggingface.co/datasets/LLM-Digital-Twin/Twin-2K-500

where $p_i(y_i)$ denotes the predicted probability assigned to the ground-truth option. Note that this differs from standard token-level language model perplexity; here PPL measures uncertainty over discrete choices.

**Brier Score (BS).** The Brier score measures the squared error between the predicted probability distribution and the one-hot encoded ground truth:

$$\text{BS} = \frac{1}{N} \sum_{i=1}^{N} \sum_{c=1}^{C} (p_{i,c} - y_{i,c})^2, \tag{12}$$

where $\mathbf{y}_i \in \{0, 1\}^C$ is the one-hot representation of $y_i$.

## C.3. Respondent Sensitivity Definition

Group-level propagation does not benefit all respondents equally: observing the ground-truth responses of some individuals yields substantially larger gains than observing others. To characterize this heterogeneity, we introduce a notion of *sensitivity*, which quantifies how informative a respondent is for downstream inference.

**Definition.** We define a respondent's sensitivity as the improvement in predictive performance attributable to observing that respondent's ground-truth responses, relative to an imputation-only baseline. Concretely, for each respondent $u$, we compute

$$\text{Sensitivity}(u) = \text{Acc}_{\text{full}}(u) - \text{Acc}_{\text{impute}}(u), \tag{13}$$

where $\text{Acc}_{\text{full}}(u)$ denotes the per-user prediction accuracy when all responses from $u$ are observed (full-observation setting), and $\text{Acc}_{\text{impute}}(u)$ denotes the accuracy when none of $u$'s responses are observed and predictions rely solely on group-level imputation. This difference captures the *marginal utility* of observing respondent $u$ beyond what can be inferred from others.

**Estimation protocol.** Sensitivity is estimated using an independently trained LLM-based predictor. For each dataset and random seed, we evaluate per-user accuracies under two controlled settings: (i) a *full-observation* setting, in which all responses from a given user are revealed, and (ii) an *imputation-only* setting, in which the same user is entirely unobserved. We compute the accuracy gap for each user and average results across multiple random seeds to reduce variance. Users are then ranked by their sensitivity scores in descending order.

**Sensitivity tiers and Interpretation.** To facilitate analysis, we stratify respondents into sensitivity tiers based on percentile ranks of the accuracy gap distribution. Specifically, we consider the top 50%, 30%, 10%, and 5% most sensitive respondents, corresponding to users for whom observing ground-truth responses yields the largest performance improvements. These tiers allow us to study where adaptive elicitation and group-relational respondent selection provide the greatest benefits, and to contrast performance on highly sensitive respondents against the broader group. Intuitively, high-sensitivity respondents are those whose preferences are poorly captured by group-level propagation alone—either because they are atypical, weakly connected, or exhibit high uncertainty—such that direct observation substantially improves inference. In contrast, low-sensitivity respondents are well-explained by relational structure and contribute limited additional information when observed. This stratification enables a more fine-grained evaluation of adaptive elicitation strategies beyond aggregate accuracy.

## C.4. Implementation Details

All experiments were conducted on a single machine equipped with 8 NVIDIA RTX 6000 Ada Generation GPUs (48 GB memory each) using CUDA 12.9.

**LLM Meta-Training.** We split the training data by entity at the respondent level, using an 80%/20% split for training and validation in the in-distribution region (South), while reserving all users from the out-of-distribution region (West) exclusively for testing. To meta-train our query selection model, we initialize a pretrained Llama-3.1-8B model and fine-tune it using parameter-efficient adaptation with LoRA, with rank $r = 8$, scaling factor $\alpha = 24$, and dropout rate 0.1, applied to the attention projection layers. Optimization is performed using AdamW with learning rate $1 \times 10^{-4}$, $\beta = (0.9, 0.95)$, and weight decay 0.1. Training is conducted in mixed precision (BF16) with gradient accumulation to achieve an effective batch size of 16, a maximum sequence length of 1024 tokens, gradient clipping at 1.0, and a cosine learning-rate schedule with linear warmup. The model is trained for 10,000 iterations with the final checkpoint selected based on the lowest

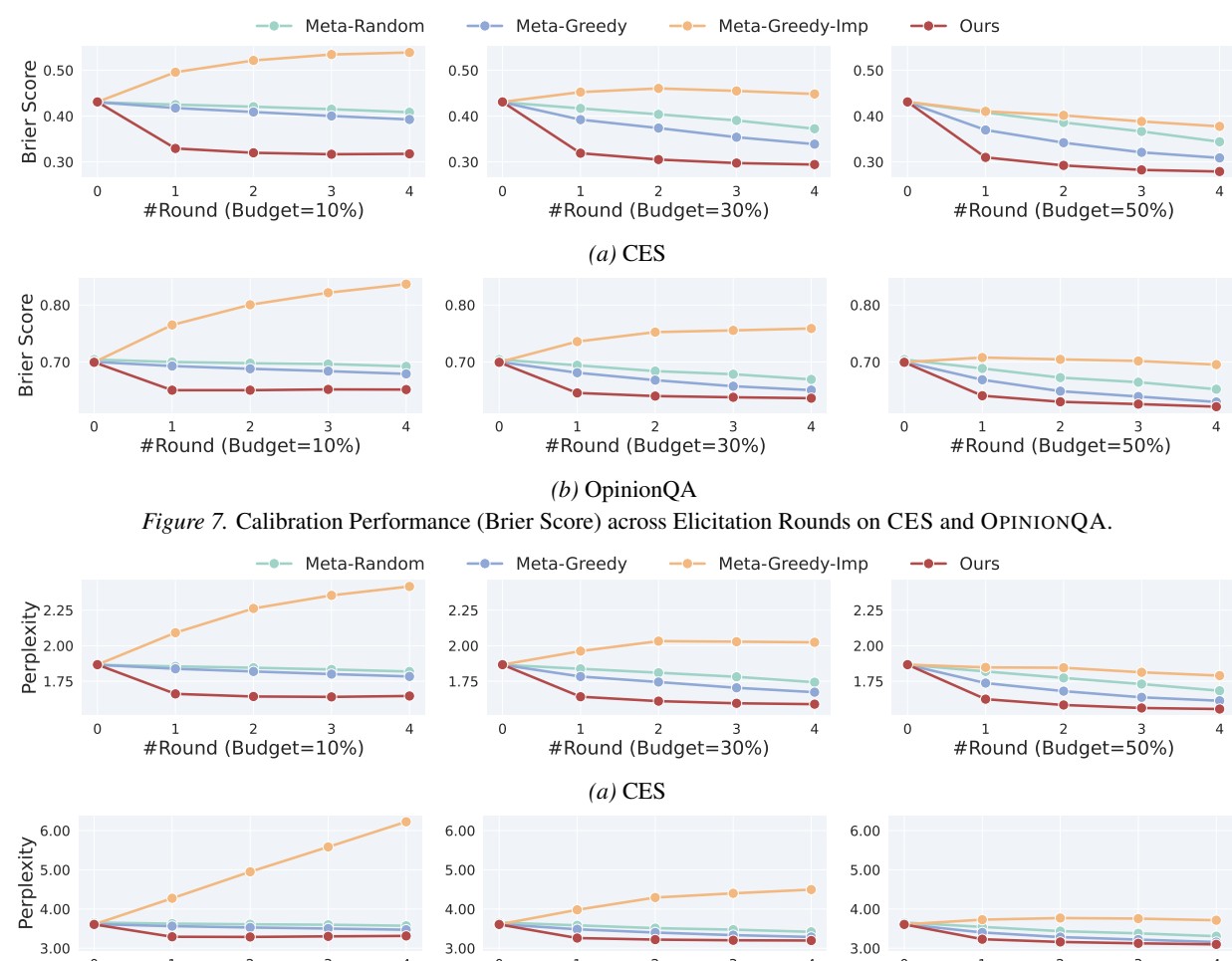

*Figure 7.* Calibration Performance (Brier Score) across Elicitation Rounds on CES and OPINIONQA.

*Figure 8.* Calibration Performance (Perplexity) across Elicitation Rounds on CES and OPINIONQA.

validation loss. For robustness, we additionally meta-train a smaller Llama-3.2-1B model under the same framework, using full-parameter fine-tuning, batch size 8, and a reduced weight decay of 0.01, while keeping the learning rate, optimizer settings, sequence length, precision, scheduling strategy, and evaluation protocol unchanged.

**Heterogeneous GNN.** We train the GNN imputation module on a heterogeneous opinion graph comprising three node types: *group member* nodes (respondents), *feature* nodes (demographic), and *query–choice* nodes (answer options). To prevent information leakage, all splits are performed at the user level, assigning all responses from a respondent to the same split. For geographic generalization, region-based filtering is applied prior to splitting, with the South region treated as in-distribution and the West region as out-of-distribution (OOD). Within the in-distribution region, users are randomly partitioned into training and validation sets with an 80/20 split using a fixed random seed, while all users from the OOD region are held out exclusively for testing. During training, we simulate partial observability via an epoch-wise user-level masking protocol: in each epoch, training users are divided into three disjoint groups, including fully observed users whose edges are retained for message passing, partially observed users for whom a fixed fraction (50%) of edges are masked and used as supervision targets, and cold-start users whose edges are entirely withheld for supervision. Validation and test splits are evaluated under a strict cold-start setting, where all user–query edges are removed from the message-passing graph. Each supervision instance corresponds to a multi-class classification problem over the candidate option set of the associated query. Unless otherwise specified, we use a hidden dimension of 64, two RGCN layers, dropout rate 0.1, batch size 2048, learning rate $2 \times 10^{-3}$ with weight decay $10^{-4}$, and train for 500 epochs with a fixed random seed.

*Table 2.* CES: Accuracy across sensitivity tiers.

| Budget | Method (Top %) | Global (100%) | Broad (50%) | Inter. (30%) | Hard (10%) | Extreme (5%) |
|---|---|---|---|---|---|---|
| 0% | Full Imp. | 0.760 | 0.705 | 0.635 | 0.477 | 0.420 |
| 30% | Random | 0.784 | 0.773 | 0.737 | 0.641 | 0.617 |
| | **Group-relational** | **0.790** | **0.782** | **0.751** | **0.684** | **0.690** |
| 50% | Random | 0.793 | 0.795 | 0.770 | 0.714 | 0.720 |
| | **Group-relational** | **0.801** | **0.812** | **0.798** | **0.780** | **0.826** |
| 100% | Full Obs. | 0.806 | 0.842 | 0.844 | 0.861 | 0.909 |

*Table 3.* OPINIONQA: Accuracy across sensitivity tiers.

| Budget | Method (Top %) | Global (100%) | Broad (50%) | Inter. (30%) | Hard (10%) | Extreme (5%) |
|---|---|---|---|---|---|---|
| 0% | Full Imp. | 0.467 | 0.383 | 0.308 | 0.220 | 0.168 |
| 30% | Random | 0.480 | 0.451 | 0.416 | 0.398 | 0.377 |
| | **Group-relational** | **0.484** | **0.460** | **0.432** | **0.430** | **0.415** |
| 50% | Random | 0.490 | 0.487 | 0.472 | 0.473 | 0.465 |
| | **Group-relational** | **0.496** | **0.496** | **0.485** | **0.522** | **0.529** |
| 100% | Full Obs. | 0.507 | 0.570 | 0.594 | 0.687 | 0.720 |

# D. Additional Experiment Results

## D.1. Calibration Results

Figure 7 and 8 reports calibration results on CES and OPINIONQA, measured by Brier Score and Perplexity as a function of elicitation rounds under different observation budgets (10%, 30%, and 50%). Lower values indicate better calibration. Across both datasets and all budgets, our method consistently achieves the lowest Brier scores and perplexities after the first elicitation round, and the gap widens as more rounds are performed. This indicates that adaptive elicitation not only improves point accuracy but also produces better-calibrated predictive distributions. In contrast, *Meta-Greedy-Imp*, which relies on LLM-independent imputation without group-relational respondent selection or propagation, often exhibits degraded calibration as the number of elicitation rounds increases, particularly under low-budget settings. This degradation is most pronounced in perplexity, indicating overconfident predictions arising from the propagation of uncertain or weakly informative signals. *Meta-Random* and *Meta-Greedy* show modest but consistent improvements with additional rounds, but remain substantially worse than our approach.

Overall, these results demonstrate that group-relational respondent selection is crucial for calibration: observing highly informative respondents early leads to uncertainty reduction that propagates reliably across the population, whereas indiscriminate propagation can amplify miscalibrated beliefs. The trends are consistent across datasets and budgets, confirming that the calibration gains of our method are robust and not an artifact of a specific domain.

## D.2. Ablation on Respondent Selection

This subsection provides supplementary ablation results for Section 5.3, focusing on how respondent selection interacts with sensitivity tiers under different observation budgets. While the main text emphasizes relative recovery to highlight where adaptive elicitation gains concentrate, here we report absolute accuracy at the final elicitation round to give a more complete picture of performance across the population.

Tables 2 and 3 report round-4 accuracy on CES and OPINIONQA, respectively, stratified by respondent sensitivity. Sensitivity tiers are defined based on the accuracy gap between full observation and imputation-only settings (Appendix C.3), ranging from *Global* (all respondents) to increasingly restrictive subsets of high-sensitivity respondents (*Broad*, *Intermediate*, *Hard*, and *Extreme*). We compare Group group-relational respondent selection against a random baseline at fixed budgets. Across both datasets, Group-relational respondent selection consistently outperforms random selection at the same budget, with gains that become more pronounced as sensitivity increases. In particular, under a 50% budget, the largest absolute improvements are observed in the *Hard* and *Extreme* tiers, where group-relational selection recovers a substantial fraction of the full-observation performance. In contrast, gains in the Global or Broad tiers are smaller, reflecting the fact that many low-sensitivity respondents are already well explained by group-level propagation. These ablations support the main conclusion of Section 5.3: respondent selection is most effective when targeted toward highly sensitive (hard) individuals, and its benefits cannot be attributed solely to increased observation volume. Instead, selecting the *right* respondents is crucial for translating limited budgets into meaningful accuracy gains.

## D.3. Results with Different Base Model

We evaluate the robustness of our approach to the choice of base language model by repeating the main elicitation experiments using a smaller backbone, LLAMA-3.2-1B, trained with full fine-tuning. Figure 9 reports accuracy across interaction rounds under different query budgets on CES and OPINIONQA. Despite the substantially reduced model capacity, our method consistently outperforms all baselines across datasets, budgets, and rounds, and exhibits qualitative trends similar to those observed with the LLAMA-3.1-8B backbone fine-tuned with LoRA. In particular, our approach achieves rapid gains in

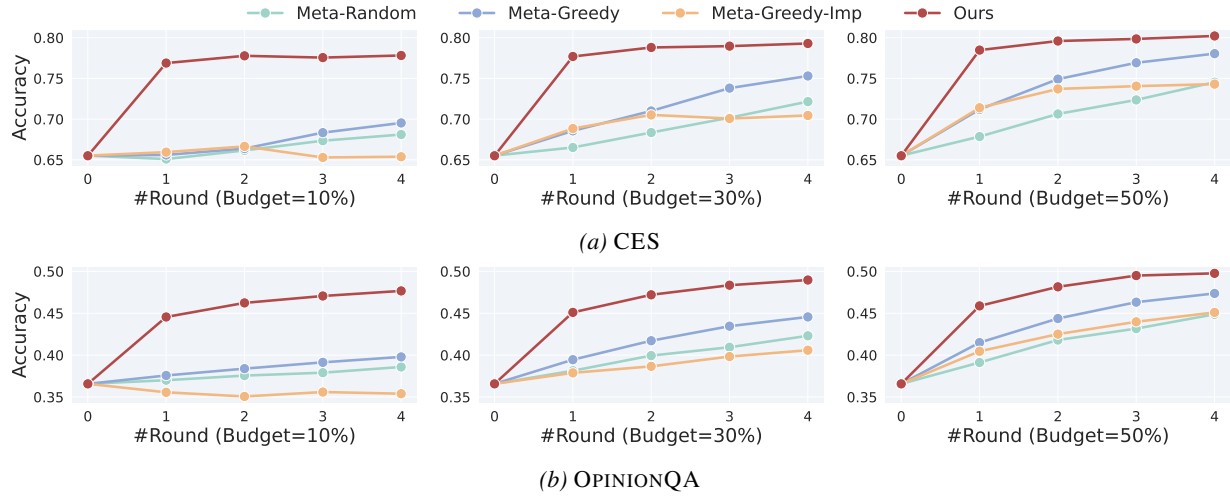

*(a)* CES

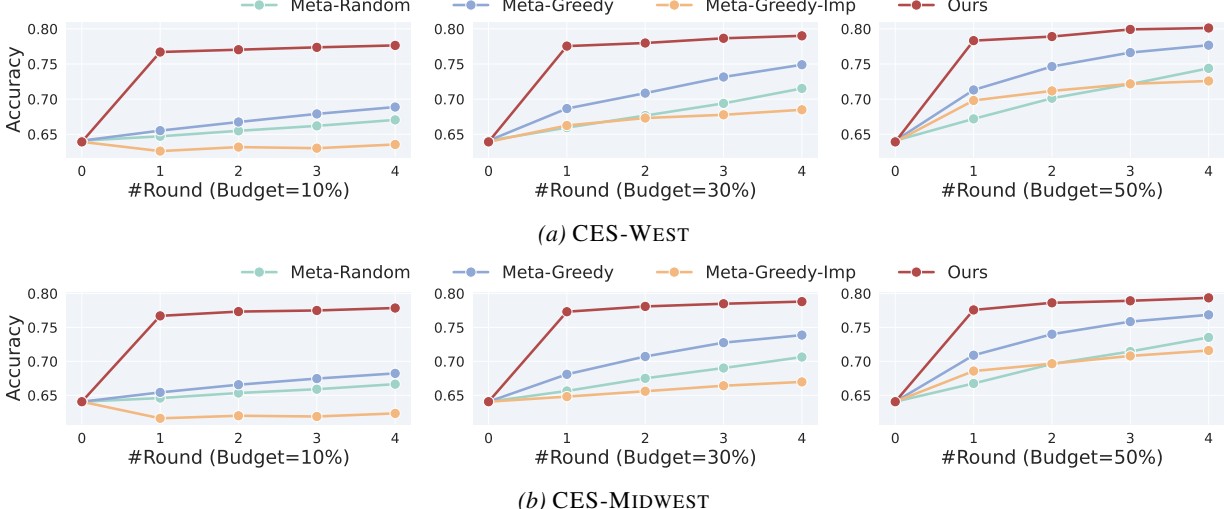

*(b)* OPINIONQA

*Figure 9.* Accuracy across interaction rounds under different query budgets using the Llama-3.2-1B backbone.

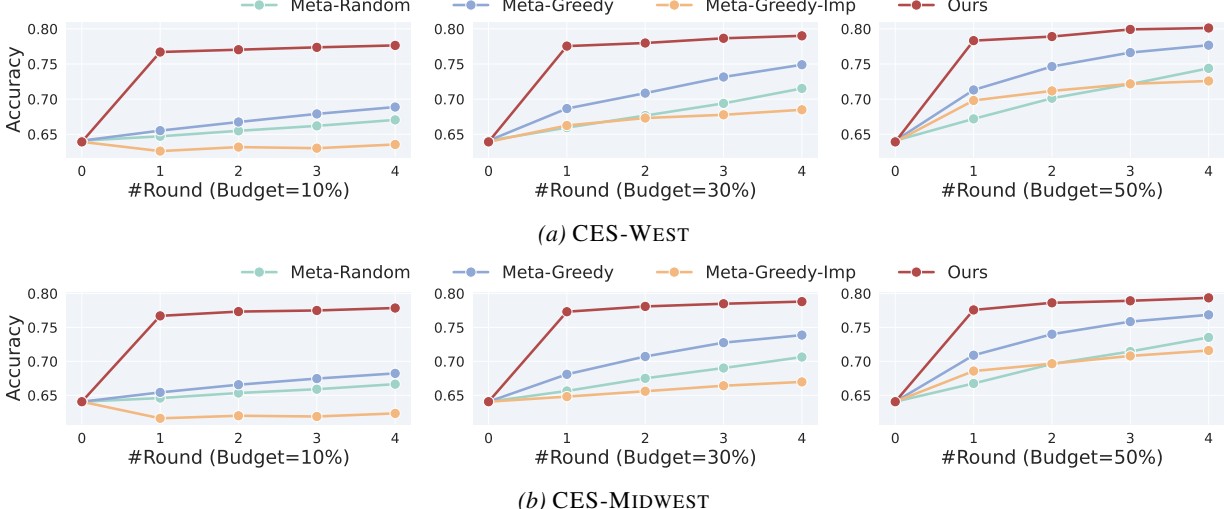

*(a)* CES-WEST

*(b)* CES-MIDWEST

*Figure 10.* Accuracy across interaction rounds under different query budgets on CES across regions.

early rounds, especially under low-budget settings, indicating that group-relational respondent selection and group-relational propagation remain effective even when the underlying LLM is smaller. Notably, we observe that full fine-tuning of the LLAMA-3.2-1B model yields performance generally comparable to LoRA-based fine-tuning of the LLAMA-3.1-8B model. These results indicate that the benefits of adaptive elicitation are largely attributable to the elicitation and propagation mechanism itself, rather than reliance on a highly capable backbone, and that the framework remains effective across different model scales and training regimes.

## D.4. Results with Different Region

We further evaluate the robustness of adaptive elicitation under geographic distribution shift by testing the model on regions not used during meta-training. Following the same experimental setup as in Section 5.2, elicitation policies are learned using respondents from the South region and evaluated on held-out regions, including the West and Midwest. Figure 10 reports accuracy across interaction rounds under different query budgets on CES for these regions. Across both regions, our method consistently outperforms baseline approaches and exhibits qualitative trends similar to those observed in the in-distribution setting. In particular, adaptive elicitation achieves substantial gains in the early rounds, even under low-budget constraints, indicating that group-relational respondent selection and group-relational propagation transfer effectively across regions. While absolute accuracies vary due to regional differences in demographic composition and opinion distributions, the relative improvements over *Meta-Random*, *Meta-Greedy*, and *Meta-Greedy-Imp* remain stable. These results suggest that the proposed elicitation framework generalizes beyond the region used for meta-training and remains effective under realistic geographic distribution shifts.

