# OpenReview forum: "Whom to Query for What: Adaptive Group Elicitation via Multi-Turn LLM Interactions"
_ICML.cc/2026/Conference — ICML 2026 regular_

### Official Review · Reviewer_BQVD · 2026-03-07

**Soundness:** 2
**Presentation:** 3
**Significance:** 2
**Originality:** 2
**Overall Recommendation:** 4
**Confidence:** 3

**Summary:**

This paper studies the adaptive group elicitation problem to adaptively select which questions to ask and which respondents to query. The proposed framework combines two components: 1) a meta-trained LLM that performs predictive inference at the individual level and scores candidate questions by expected information gain; and 2) a heterogeneous GNN that propagates information across the group to impute missing data. Experiments on three real-world opinion datasets show consistent performance improvements from the proposed method.

**Compliance With Llm Reviewing Policy:**

Affirmed.

**Final Justification:**

My concerns have been addressed, and the scores have been accordingly updated.

**Key Questions For Authors:**

1. Please briefly discuss the potential limitations of the proposed framework, e.g., when the method/assumptions may fail, the scalability, the reliance on demographic features, etc.
2. Does the error propagation from inaccurate imputations degrade performance over rounds?
3. Please discuss more on the generalisation problems mentioned in Weaknesses.

**Limitations:**

Please briefly discuss the potential limitations of the proposed framework, e.g., when the method/assumptions may fail, the scalability, the reliance on demographic features, etc.

**Strengths And Weaknesses:**

Strengths:
1. The research question this paper focuses on, i.e., how to optimise what to ask and who to ask, is interesting and important in adaptive elicitation literature. It reflects common survey constraints in practice.
2. This paper provides near-optimality guarantees for the greedy and two-stage greedy algorithms. The proofs in the appendix are detailed and appear technically sound.
3. Experiments on three datasets with multiple budget levels show the effectiveness of the proposed method. Ablation studies further verify the contribution of query selection, GNN-based imputation, and respondent selection.

Weaknesses:
1. Both Theorems 4.1 and 4.2 rely on Assumption B.1. However, it does not provide verification, either empirical or theoretical, on whether the devised function satisfies both monotonicity and submodularity in practice.
2. At test time, the direct concatenation of imputations from GNN may introduce imputation noise and interfere with LLM's uncertainty estimates. It is suggested to discuss whether error propagation from inaccurate imputations degrades performance over rounds.
3. All three datasets are opinion surveys with multiple-choice responses from the US population. While the paper motivates broadly, e.g., student assessments, employee preferences, the evaluation is limited to political or social opinion data. It leads to the question of whether the proposed method can transfer well to other settings, e.g., with continuous responses or domains where the latent entity has different characteristics.
4. At test time, the candidate query and evaluation set are drawn from the same pool of questions used during training. But in practice, new survey questions may be introduced that weren't in the training set. Then, how does the LLM handle novel question content, and since the LLM is fine-tuned on a fixed set of policy questions, is it able to generalise to unseen topics?
5. The computational cost analysis is missing. It is suggested to provide the theoretical or empirical comparison of computational cost and discuss the scalability of the proposed framework for practical survey deployment.
6. It is also suggested to provide some case studies to illustrate and analyse the success and failure cases.

---

> ### Author Rebuttal · Authors · 2026-03-31
>
> ### W1: Verification for Monotonicity and Submodularity Assumptions
>
> We thank the reviewer for this question and verify Assumption B.1 below in both theory and experiments.
>
> **Theory:** Our group Expected Information Gain (EIG) is the average of individual EIGs. Since both properties are preserved under summation, it is sufficient to prove them for single member EIG. Denote $W_v$ the set of queries assigned to member $v$, and $w$ a new query.
>
> - *Monotonicity*: We have $H(U_v|Y^{W_v \cup \{w\}}) \le H(U_v|Y^{W_v})$, taking expectation over $Y$ gives monotonicity of EIG.
>
> - *Submodularity*: Individual EIG equals to $H(Y^{W_v})-H(Y^{W_v} | U_v)$. Assuming responses are independent condition on $U_v$, the second term becomes $\sum_{x \in W_v} H(Y^x | U_v)$ and is modular. The first term is submodular since conditioning on a larger set of responses reduces the information gain of a new response.
>
> **Estimation:** While true EIG objective obeys these laws, practical estimation involves approximation using GNN imputations and LLM predictive inference, thus these properties remain theoretical assumptions.
>
> **Experiment: empirical trends validate the assumptions.**
> We observe a **non-decreasing cumulative EIG** and **diminishing marginal gains** across rounds. Cumulative EIG increases 0.00 → 0.18, while marginal gain drops 0.0355 → 0.0015 (first 5 rounds). These two signatures directly validate the assumptions:
> - *Monotonicity*: cumulative EIG should not decrease with more response data.
> - *Submodularity*: marginal gains should decrease as the selected set grows.
>
> The strong alignment between observed trends and these theoretical signatures provides **empirical validation that our objective satisfies monotonicity and submodularity in practice**, despite estimation noise. See the full **[results and plots](https://anonymous.4open.science/r/ICML-GAE-Rebuttal-BDB4/BQVD/W1.md)**.
> |R|Cumulative EIG (↑)|Mean ΔEIG (↓)|
> |-|-|-|
> |1|0.0000|0.0355|
> |3|0.1752|0.0020|
> |5|0.1823|0.0015|
>
> ---
>
> ### W2: Error Propagation from Imputation
>
> **Imputation is imperfect but does not cause error accumulation.** Despite declining imputation accuracy (0.852→0.712), LLM accuracy improves (0.778→0.807) (**[Table](https://anonymous.4open.science/r/ICML-GAE-Rebuttal-BDB4/BQVD/W2.md)**).
> **Reason: signal dominates noise.**  Iterative querying expands informative coverage, and structure-aware imputations remain useful. **Active querying further corrects errors.** Ours > Ours-Imp (**Fig 6, line 410**), showing observations provide additional corrective signal.
>
> ---
>
> ### W3: Generalization Beyond Opinion Data
>
> We thank the reviewer for highlighting the importance of broader evaluation. **Our framework is domain-agnostic.** We evaluate on political opinion (CES, OpinionQA) and economic preference (Twin-2K), and extend to any setting with latent user state and discrete responses. **We include an additional non-opinion task.** On a cognitive–metacognitive dataset (DUNNING–KRUGER, **[link](https://anonymous.4open.science/r/ICML-GAE-Rebuttal-BDB4/BQVD/W3.md))**, we observe consistent gains (R1–R4: 0.46→0.54). We will expand evaluation and welcome suggestions on relevant datasets.
>
> ---
>
> ### W4: Generalization to Unseen Questions and Novel Query Distributions
>
> **Ours generalizes beyond seen questions with recoverable performance gaps.** We split CES into 15 train / 5 unseen questions. While unseen questions show an initial drop (R1: 0.79→0.69), performance improves over rounds (R1-R4: 0.69→0.75). Interaction mitigates question shift. **[See results](https://anonymous.4open.science/r/ICML-GAE-Rebuttal-BDB4/BQVD/W4.md)**.
>
> ---
>
> ### W5: Computational Cost Analysis and Scalability
>
> Our method has comparable cost to baselines and scales efficiently. Please refer to our response to **Reviewer ZYYd (W2)** for details.
>
> ---
>
> ### W6: Qualitative Case Studies
>
> **Ours gains from semantically aligned queries but can fail due to imputation errors.**
> Ours selects high-relevance queries (e.g., policing→gun), achieving faster gains (0.782→0.835 vs. Random 0.732→0.764; [Table](https://anonymous.4open.science/r/ICML-GAE-Rebuttal-BDB4/BQVD/W6.md)). Failures arise from incorrect imputations—e.g., a wrong imputation on a key query flips the final prediction ([Table](https://anonymous.4open.science/r/ICML-GAE-Rebuttal-BDB4/BQVD/W6.md)).
>
> ---
>
> ### Q1: Limitations
>
> Two main limitations: reliance on pretraining priors and lack of formal error control. The LLM predictor may degrade on underrepresented or unseen subpopulations, and while GNN imputation is effective, it introduces noise without formal guarantees (e.g., prediction-powered inference bounds).
>
> We will expand the discussion of limitations in the paper.
>
> ---
>
> ### Q2: Imputation Errors
>
> Please refer to W2.
>
> ---
>
> ### Q3: Generalization Limitations
>
> Please refer to:
> - **W3:** generalization beyond opinion data
> - **W4:** generalization to unseen questions

---

> > ### Author Rebuttal · Reviewer_BQVD · 2026-04-02
> >
> > Thank you for your rebuttal. I will consider raising my score.

---

> > > ### Author Response · Authors · 2026-04-04
> > >
> > > Thank you very much for your positive reassessment. We sincerely appreciate the time and care you took to review our revisions, and we are glad that the additional experiments and clarifications helped address your concerns.
> > >
> > > We noticed your comment about considering raising your score, and we would be truly grateful if you would update it in the system to reflect your revised evaluation.
> > >
> > > We also want to assure you that we will carefully incorporate all the improvements discussed into the final version. Thank you again for your thoughtful feedback and consideration.

---

### Official Review · Reviewer_ZYYd · 2026-03-10

**Soundness:** 3
**Presentation:** 3
**Significance:** 4
**Originality:** 3
**Overall Recommendation:** 4
**Confidence:** 3

**Summary:**

This paper studies the problem of adaptive information elicitation for inferring latent group-level properties under limited query and respondent budgets. The authors propose a framework in which a central agent interacts with a population over multiple rounds, adaptively selecting both the next query and the subset of respondents to maximize information gain about a latent population attribute

**Compliance With Llm Reviewing Policy:**

Affirmed.

**Key Questions For Authors:**

How does the framework perform when demographic features are sparse or weakly predictive of opinions?

**Limitations:**

yes

**Strengths And Weaknesses:**

Strengths: clear problem formulation and reasoning, interesting hybrid architecture LLM + GNN, the experimental evaluation uses three real-world datasets with demographic attributes and opinion questions + additional ablations. This paper has clear applications for social scientists like survey design, public opinion research and adaptive questionnaires.

Weaknesses: the evaluation measures performance mainly through prediction accuracy on held-out responses. While appropriate, the paper could benefit from additional analyses such as: robustness to demographic distribution shifts, sensitivity to graph construction choices and computational cost comparisons with simpler baselines. The interaction process is simulated using existing survey datasets rather than real adaptive interactions with participants. While this is common in adaptive survey research, it leaves open questions about how the system performs in real deployment settings with: non-response behavior, noisy responses, changing participant pools.

---

> ### Author Rebuttal · Authors · 2026-03-31
>
> We sincerely thank the reviewer for the positive and constructive suggestions.
> **Prediction accuracy is a principled metric for our objective.** Our goal is to recover latent group-level preferences from partial observations; under exchangeability assumptions, accurate prediction of held-out responses reflects correct recovery of the underlying population latent opinions' distribution. This connection is established via a deep theoretical result using De Finetti's Theorem. We would like to point out, **we indeed also report Perplexity and Brier Score in Appx E.1, confirming improved probabilistic calibration beyond accuracy**.
>
> ***To further ease your concern, we also include the suggested additional analyses below.***
>
> ### W1: Robustness to Demographic Distribution Shift
>
> **Our method generalizes to unseen regions.** We train on **South** and evaluate on **West/Midwest** (see App. E.4, Fig. 10). The advantage is consistent and strongest early: at Round 1 (10%), ~0.77 vs ~0.65 (best baseline) in both regions, with gains persisting across budgets (30%, 50%) and rounds. This demonstrates robustness under distribution shift, with trends closely matching in-distribution results.
>
> ---
>
> ### W2: Computational Cost Comparison
>
> **Our method has comparable inference cost to Meta-Greedy Baseline, with negligible GNN overhead.**
>
> - *LLM EIG Estimation (Dominant):* Computing EIG requires sampling simulated future responses to evaluate conditional entropy. The algorithm is of complexity $\mathcal{O}(|\mathcal{V}| \times |{\mathcal{X}_{c}}| \times N \times |{\mathcal{X}_h|})$, where $\mathcal{V}$ is the test-time group scale,  X_c, X_h represent the candidate and held-out query sets, and $N$ is the sampling number. Our method and all baselines involving strategic query selection share this exact LLM sampling bottleneck, random query selection baseline bypasses this complexity at the cost of downstream performance.
>
> - *GNN Imputation (Negligible):* The GNN formulates imputation as link prediction over a heterogeneous graph. The graph contains three node types (group members $\mathcal{V}$, demographic features $\mathcal{V}_f$, discrete queries $\mathcal{V}_c$) and two edge types (member-feature edges $\mathcal{E}_f$, and observed member-choice responses edges $\mathcal{E}_c$). Its message-passing complexity is proportional to the graph size: $\mathcal{O}(|\mathcal{V}| + |\mathcal{V}_f| + |\mathcal{V}_c| + |\mathcal{E}_f| + |\mathcal{E}_c|)$. Because the interaction query budget is small, the observed response edges $|\mathcal{E}_c|$ remain highly sparse, making this step efficient.
>
> **Empirically, runtime matches baselines.**  Over 4 rounds on 4×A6000: 6.2 min (1B) / 30.4 min (8B),  comparable to Meta-Greedy (same EIG), with negligible GNN overhead.
>
> **Training is a one-time offline cost.**  LoRA (8B, 10k steps) takes ~6.7h (26.7 GPU-hours), comparable to simpler baselines.
>
> **Better sample efficiency improves overall cost.**  We achieve target accuracy with fewer queried respondents, reducing real-world data collection cost.
>
> ---
>
> ### W3: Simulated vs Real-World Deployment
>
> We thank the reviewer for raising these important aspects of real-world deployment.
>
> **Our partial observation setting closely mirrors real-world non-response.** We evaluate under 10% / 30% / 50% observation (i.e., up to **90% missing**), as shown in Fig. 3 (line 349), which closely reflects realistic non-response regimes. Missing entries are naturally represented as **absent edges** and handled via GNN-based imputation.
>
> **We appreciate the suggestion on noisy responses.** We will include additional experiments with noisy/corrupted responses to further evaluate robustness.
>
> **Changing participant pools are already considered.** As discussed in **W1 (Robustness to Demographic Distribution Shift)**, our method generalizes across regions, demonstrating robustness to shifting populations.
>
> ---
>
> ### Q1: Sparse Demographic Features
>
> **Our framework remains robust when demographic features are sparse.** We simulate weak demographic signals by independently masking each demographic attribute with probability (10%,30%,50%), while keeping all other components unchanged.
>
> **Performance drops moderately, mainly in early rounds, but remains clearly above the strongest baseline.** As shown in this **[link](https://anonymous.4open.science/r/ICML-GAE-Rebuttal-BDB4/ZYYd/Q1.md)**, the largest degradation occurs at Round 1, where 50% masking reduces accuracy from 0.7912 to 0.7520. However, this gap quickly narrows over rounds. Even with 50% missing demographics, our method still consistently outperforms Meta Greedy, e.g., 0.7520 vs. 0.7155 at Round 1 and 0.7826 vs. 0.7716 at Round 4. **This suggests the framework does not rely heavily on demographic priors.** Instead, it can recover informative signals through observed responses, graph structure, and adaptive querying, making it robust when demographic features are sparse or weakly predictive.

---

### Official Review · Reviewer_b91v · 2026-03-16

**Soundness:** 3
**Presentation:** 3
**Significance:** 3
**Originality:** 4
**Overall Recommendation:** 5
**Confidence:** 3

**Summary:**

This paper proposes a framework for adaptive group elicitation, where the language model delibrates on what questions should be asked and to which virtual user the questions should be asked. The core idea combines a fine-tuned LLM (used to estimate expected information gain (EIG) and select maximally informative questions) with a graph neural network, modeling relational structure among respondents and impute missing answers. At each round, the system picks the most informative question and a diverse, representative subset of respondents via embedding-space clustering, then propagates observed responses across the group graph to fill in unobserved ones. Experiments on three opinion datasets (CES, OpinionQA, TWIN-2K) show consistent accuracy improvements over baselines, with a headline >12% relative gain on CES at a 10% respondent budget.

**Compliance With Llm Reviewing Policy:**

Affirmed.

**Final Justification:**

**Post-Rebuttal**:
Authors have provided additional results and explanations that help resolve the said weaknesses and questions.
Accordingly updating my evaluation of the submission.

**Key Questions For Authors:**

* Do you see large differences in EIG across different query choices?

**Limitations:**

Yes

**Strengths And Weaknesses:**

Strengths:
* Proposed methdology of adaptive group elicitation (building on existing work) is well motivated and clearly articulated.
* Experiments and results from ablation studies attempt at and help decomposing gains from each component of the proposed method.

Weaknesses:
* While the problem motivation is understandable, the evaluation setup could introduce more explicitly LLM-based baselines: how does the proposed method compared to an entirely LLM-based method, given that the same information about users and questions are provided? More details on the existing baseline methods (what prompt, how exactly does inference work in your evaluation setup work for the baselines) will help reader understanding.
* Authors report that results are based on "20 candidate questions and 5 target questions": more information on how the target questions are selected (i.e. randomly, each time for the 10 trials?) and why only 5 target questions were used for evaluation (e.g. the used datasets definitely have more survey questions) would be necessary
* Is there any validation on the subgroup clustering based on GNN-embeddings? For example, it would be good to see some statistic on cluster scatter, to see if the GNN embeddings provide meaningful signal for clustering.
* Why are the evaluations performed up to only four rounds of interaction? Figure 3 show that while the proposed method largely plateaus after the first round, baseline methods continue to improve in accuracy over rounds, and with more rounds of interaction we might see baselines out-performing the proposed method.

---

**Post-Rebuttal**:
Authors have provided additional results and explanations that help resolve the said weaknesses and questions.
Accordingly updating my evaluation of the submission.

---

> ### Author Rebuttal · Authors · 2026-03-31
>
> ### W1: Comparison with Fully LLM-Based Baselines and Details
>
> We thanks the reviewer for the comment, but we want to clarify that **our setup indeed already included fully LLM-based baselines (see line 295).**
> The Meta-* methods operate directly with a pretrained LLM at inference time, including:
> (i) Meta-Random (LLM-only prediction),
> (ii) Meta-Greedy (LLM-based EIG query selection),
> (iii) Meta-Greedy-Imp (fully LLM-based querying + imputation).
>
> **All methods share the same interactive protocol for fair comparison**. Each round follows the same pipeline: query selection → partial observation → (optional) imputation → prediction under identical budgets.
>
> **We also include additional off-the-shelf LLM baselines.** Instruct-InContext directly uses demographic and interaction context without meta-training. **Our method consistently outperforms all LLM-only baselines.** On CES (Round 4): Base-Random 0.474, Instruct-InContext 0.523, Meta-Greedy 0.749, Meta-Greedy-Imp 0.685, **Ours 0.790 (+4.1 over Meta-Greedy)**. Similar trends hold on OpinionQA (0.484 vs 0.431 best baseline), showing that *LLM-only methods saturate while structured modeling brings additional gains*.
>
> For detailed results, see the following **[link](https://anonymous.4open.science/r/ICML-GAE-Rebuttal-BDB4/b91v/W1.md)**.
>
>
>
>
> ---
>
> ### W2: Target Question Selection and Evaluation Scope
>
> **Target questions are randomly re-sampled per trial for unbiased evaluation.** In each trial, we sample interaction candidates and *randomly select non-overlapping target questions*, repeated over 10 trials.
>
> **5 targets provide an efficient and robust estimate.** Instead of evaluating all questions, we use **5 targets × 10 trials** to approximate expectation over subsets with manageable cost.
>
> **We additionally include experiments with varying target sizes.** Varying target question numbers ∈ {1,3,5,10} yields minimal change for Ours (0.769–0.773), consistently outperforming Meta-Greedy (~0.73). **Results are stable across target sizes.** See details in this **[link](https://anonymous.4open.science/r/ICML-GAE-Rebuttal-BDB4/b91v/W2.md)**.
>
> **Our method are not sensitive to target design.** Performance gains remain *robust to both target size and sampling*.
>
>
> ---
>
> ### W3: Validation of GNN-Based Subgroup Clustering Quality
>
> **GNN embeddings yield meaningful and stable subgroup structure (Silhouette ≈0.22–0.24).** We evaluate clustering quality via Silhouette (↑, typical real-world ≈0.2–0.5), obtaining 0.241 (R1), 0.238 (R3), 0.224 (R5).
>
> **Clustering quality is stable across rounds.** The small variation (0.241 → 0.224) indicates robustness as interaction proceeds.
>
>
> **Embedding Visualization aligns with metrics.** t-SNE results shows clusters form compact, well-separated groups that remain consistent across rounds, suggesting they capture stable and meaningful subpopulation structure rather than random partitioning. See link: **[t-SNE Visualization](https://anonymous.4open.science/r/ICML-GAE-Rebuttal-BDB4/b91v/clue_cluster_overview.png)**.
>
> ---
>
>
> ### W4: Interaction Rounds and Performance Trends
>
> **We focus on early rounds (≤4) to reflect realistic low-budget settings.**  Each query incurs cost, so the goal is **maximizing accuracy under limited interaction**, not asymptotic performance.
>
> **Our method achieves clear gains in early rounds (key regime).**
> Round 1: 0.7912 vs. 0.7155 (+7.6%);
> Round 2: 0.8119 vs. 0.7356 (+7.6%);
> Round 4: 0.8213 vs. 0.7716.
> This highlights **superior sample efficiency**.
>
> **With more rounds, baselines improve but never surpass ours.** Over 20 rounds, Meta-Greedy: **0.6784 → 0.8140**, Ours: **0.6784 → 0.8278**;  the gap narrows but *ranking remains unchanged*.
>
> **Convergence under full observation is expected.** When all questions are asked, performance saturates (~0.839) and differences diminish, so the key evaluation is **early-stage efficiency**. See details via **[link](https://anonymous.4open.science/r/ICML-GAE-Rebuttal-BDB4/b91v/W4_line_chart.png)**.
>
> ---
>
> ### Q1: Variability of EIG Across Different Query Choices
>
> **Consistent EIG gaps indicate non-trivial query choice.** On CES data, across all rounds, the top-1 query achieves 56%–67% higher EIG than the mean (R1: 0.120 vs 0.053; R4: 0.009 vs 0.003), showing a strong and stable ranking signal among candidates.
>
> **The gap persists despite diminishing absolute EIG.** While EIG values decrease over rounds (mean: 0.053 → 0.003), the relative advantage remains high (56% → 67%), indicating that selecting the right query continues to matter throughout interaction.
>
> **Query selection remains impactful at every round.** These results confirm that EIG differences across candidates are substantial and persistent, making query selection a non-trivial decision problem rather than a near-random choice.
>
> See details via **[Full Results Table](https://anonymous.4open.science/r/ICML-GAE-Rebuttal-BDB4/b91v/Q1.md)**.

---

> > ### Author Rebuttal · Reviewer_b91v · 2026-04-04
> >
> > Thank you for the explanations and additional analyses:
> >
> > ---
> >
> > W1: The extra results of comparison is helpful and resolves the said point about comparison against LLM-based baselines.
> > Inclusion of the exact prompts used in the Appendix will be great; if you can provide an example of prompt used in the Instruct-InContext baseline (with the details filled out) would be helpful.
> >
> > That is, an example of how the user profile information is formatted: this wil help readers appreciate under what prompt the results in the linked markdown table was obtained.
> > > [User Profile]
> > Age: {age}, Gender: {gender}, Region: {region}, ...
> >
> >
> > W2: The added ablations with increasing target size is helpful. It would be great to see this included as part of the manuscript (mentioned in main and table provided in appendix)
> >
> > W3: Results of the silhouette scores across rounds is helpful in understanding the clustering quality: thanks for providing these values and will help readers appreciate the proposed method.
> >
> > W4: Thank you for the full, expanded results provided in the link: it would be good to have this be presented in the main manuscript results. I do appreciate the reasoning behind why smaller number of rounds reflect low-resource constraints; provided both with the extended results, we can see how *all* methods (the presented method, baselines, as well as the idealized baseline of using all respondents) plateau after 10 rounds. Reflecting this in the results will be great for readers.
> >
> > Q1: The results on EIG across query choices is definitely helpful in understanding the value of the proposed methodology around query selection. Having this in the manuscript will reinforce the contributions made by this work.

---

> > > ### Author Response · Authors · 2026-04-06
> > >
> > > We sincerely thank you for your valuable time and effort in reviewing our manuscript and rebuttal, as well as for your insightful and constructive suggestions. We are very encouraged that our rebuttal has addressed your concerns and led to a more positive assessment. We will incorporate the corresponding revisions into the final version of the manuscript.

---

### Official Review · Reviewer_7iCk · 2026-04-06

**Soundness:** 2
**Presentation:** 2
**Significance:** 3
**Originality:** 3
**Overall Recommendation:** 4
**Confidence:** 3

**Summary:**

This paper introduces a method for selecting respondents (human subjects) from a group to efficiently querying their opinions under budget constraint, in a multi-turn LLM interactive setting. The key novelty is the creation of a heterogeneous respondent-attribute-query graph, the authors propose to learn node embeddings on this graph and use clustering results of the respondent nodes for picking respondents. Experiments show more accurate response prediction on several opinion datasets.

**Compliance With Llm Reviewing Policy:**

Affirmed.

**Final Justification:**

na

**Key Questions For Authors:**

Please see weaknesses

**Limitations:**

Yes.

**Strengths And Weaknesses:**

Strengths:
1. This paper studies an important problem of surveying respondents under budget constraint. The idea of building a heterogeneous graph to help pick respondent is novel and reasonable.
2. The objective of selecting query and respondents to minimize uncertainty is well-designed and backed by good theoretical justification.
3. Experiments are solid and show substantial improvement of prediction accuracy.
4. The overall presentation of the paper is good. Experiments are presented clearly and analysis are in-depth.


Weaknesses:
1. I am a bit confused about how the authors initialize the embeddings for new member nodes at inference time. The GNN was only trained on member nodes that exist in your training set. At inference time, you mentioned that new member nodes features are initialized "uniformly" (at random?) How would you be able to do clustering over the GNN latent embeddings generated on those node features?
2. This question is related to the previous one and may be addressed together once the initialization for member nodes is sorted out. I am not sure if the usage of GNN is necessary. It seems that the goal of learning on the constructed heterogeneous graph is simply to obtain node embeddings (that encodes node proximity information, to be useful for clustering). The GNN itself is not used as a function for transforming unseen node features? In that regard I feel that many node embedding methods could potentially work?
3. This paper would benefit more from creating a dedicated "problem definition" paragraph or section. Currently it takes a while to understand what exactly the task is, and what exact budget constraint we are subject to.

---

### Decision · Program_Chairs · 2026-04-30

**Decision:**

Accept (regular)

**Comment:**

This paper studies an important problem of surveying respondents under a budget constraint. It then proposes a hybrid LLM+GNN framework to jointly select queries and a subset of respondents under budget. The reviewers agree that the problem is important, and the approach is technically solid, with consistent empirical gains across datasets.

Concerns.
- There were some initial concerns around clarity (problem setup, evaluation protocol), the role of the GNN, and missing comparisons/analyses. The author's rebuttal addresses most of these: it clarifies inference cost, strengthens justification for the GNN (beyond static embedding methods), and adds useful experiments (LLM baselines, clustering quality, robustness, and longer rounds). Several reviewers explicitly noted their concerns were resolved.
- Some limitations remain, esp. around evaluation realism, broader generalization, and presentation clarity. But these do not undermine the core contribution, but slightly limit its impact.

Overall, it's a solid and reasonably novel contribution with convincing results, and most concerns are addressed after rebuttal.